# Microspectroscopic visualization of how biochar lifts the soil organic carbon ceiling

Zhe (Han) Weng[1,2,3,4], Lukas Van Zwieten ⬦[1,5] ✉, Ehsan Tavakkoli ⬦[6,7], Michael T. Rose[1], Bhupinder Pal Singh[2], Stephen Joseph[8,9], Lynne M. Macdonald[10], Stephen Kimber[1], Stephen Morris[1], Terry J. Rose[5], Braulio S. Archanjo ⬦[11], Caixian Tang[3], Ashley E. Franks ⬦[12,13], Hui Diao[14], Steffen Schweizer ⬦[15], Mark J. Tobin[16], Annaleise R. Klein[16], Jitraporn Vongsvivut[16], Shery L. Y. Chang[17], Peter M. Kopittke ⬦[4] & Annette Cowie ⬦[2,18]

The soil carbon (C) saturation concept suggests an upper limit to the storage of soil organic carbon (SOC). It is set by the mechanisms that protect soil organic matter from mineralization. Biochar has the capacity to protect new C, including rhizodeposits and microbial necromass. However, the decadal-scale mechanisms by which biochar influences the molecular diversity, spatial heterogeneity, and temporal changes in SOC persistence, remain unresolved. Here we show that the soil C storage ceiling of a Ferralsol under subtropical pasture was raised by a second application of *Eucalyptus saligna* biochar 8.2 years after the first application—the first application raised the soil C storage ceiling by 9.3 Mg new C ha⁻¹ and the second application raised this by another 2.3 Mg new C ha⁻¹. Linking direct visual evidence from one-, two-, and three-dimensional analyses with SOC quantification, we found high spatial heterogeneity of C functional groups that resulted in the retention of rhizodeposits and microbial necromass in microaggregates (53–250 μm) and the mineral fraction (<53 μm). Microbial C-use efficiency was concomitantly increased by lowering specific enzyme activities, contributing to the decreased mineralization of native SOC by 18%. We suggest that the SOC ceiling can be lifted using biochar in (sub)tropical grasslands globally.

[1]NSW Department of Primary Industries, Wollongbar Primary Industries Institute, Wollongbar, NSW 2477, Australia. [2]School of Environmental and Rural Science, University of New England, Armidale, NSW 2351, Australia. [3]Department of Animal, Plant & Soil Sciences, Centre for AgriBioscience, La Trobe University, Melbourne, VIC 3086, Australia. [4]School of Agriculture and Food Sciences, The University of Queensland, St. Lucia, QLD 4072, Australia. [5]Southern Cross University, East Lismore, NSW 2480, Australia. [6]NSW Department of Primary Industries, Wagga Wagga Agriculture Institute, Wagga Wagga, NSW 2650, Australia. [7]School of Agriculture, Food & Wine, The University of Adelaide, Glen Osmond SA 5064, Adelaide, Australia. [8]Institute for Superconducting and Electronic Materials and School of Physics, University of Wollongong, Wollongong, NSW 2522, Australia. [9]School of Materials Science and Engineering, University of New South Wales, Sydney, NSW 2052, Australia. [10]CSIRO Agriculture & Food, Waite campus, Glen Osmond, SA 5064, Australia. [11]Materials Metrology Division, National Institute of Metrology, Quality and Technology (INMETRO), Rio de Janeiro 25250–020, Brazil. [12]Department of Physiology, Anatomy and Microbiology, La Trobe University, Melbourne, VIC 3086, Australia. [13]Centre for Future Landscapes, La Trobe University, Melbourne, VIC 3086, Australia. [14]Centre for Microscopy and Microanalysis, The University of Queensland, Brisbane, QLD 4072, Australia. [15]School of Life Sciences, Technical University of Munich, Munich, Germany. [16]Australian Nuclear Science and Technology Organisation (ANSTO), Australian Synchrotron, Clayton, VIC 3168, Australia. [17]Electron Microscope Unit, Mark Wainwright Analytical Centre and School of Materials Science and Engineering, University of New South Wales, Sydney, NSW 2052, Australia. [18]NSW Department of Primary Industries, Armidale, NSW 2351, Australia. ✉e-mail: lukas.van.zwieten@dpi.nsw.gov.au

Human activities risk releasing 260 Pg of ecosystem carbon (C) as carbon dioxide ($CO_2$) globally that is irrecoverable on a timescale relevant to avoiding profound climate impacts[1,2]. Agriculture contributes a major part, releasing an average of 2 Mg C ha$^{-1}$ y$^{-1}$ from soil globally[3–5]. Plants release ~50% of photosynthetically fixed C into the soil, which supports microbial growth and metabolism, including respiration that produces $CO_2$. It has been estimated that 122 Mg soil organic C (SOC) ha$^{-1}$ to a depth of 1 m has been lost over 1 Mha of land converted to tropical grasslands[6], with 40% of this area occurring on Ferralsols[7]. The grand challenge humanity now faces is to urgently reverse this loss of SOC and associated decline in soil health by increasing the amount of C retained in soil[5,8,9].

The Intergovernmental Panel on Climate Change (IPCC) has identified that substantial $CO_2$ removal will be required to limit global warming to 2 °C. To this end, the IPCC has identified soil C management[4–6] and the application of biochar[10] as carbon dioxide removal (CDR) methods[11] with considerable potential, with corollary benefits of improving soil health, sustaining agricultural productivity[12,13], and increasing the resilience of ecosystem services[14,15]. Protecting and rebuilding soil C could sequester 5.5 Pg $CO_2$ y$^{-1}$, representing 25% of the potential of natural climate solutions to deliver CDR through conservation, restoration, and improved land management practices[6,16,17].

Application of biochar is a recognized CDR method because of its persistence[9,11] in the environment. The pyrolysis of biomass can deliver bioenergy, as well as agronomic and non-$CO_2$ greenhouse gas benefits through the use of biochar as a soil amendment[18–22]. Biochar systems generally show life-cycle climate change impacts of net emission reduction in the range of 0.4–1.2 Mg $CO_2$ equivalent Mg$^{-1}$ dry feedstock[23], through C persistence and avoided non-$CO_2$ emissions. The capacity for biochar to further contribute to mitigation by protecting and building SOC is often overlooked.

Here, we assess the capacity and mechanisms by which biochar builds new biogenic SOC reserves. We propose a mechanism by which biochar accelerates the formation of microscale organo-mineral and nanoscale organo-organic interfaces in soil microaggregates (53–250 μm) and mineral fractions (<53 μm) to protect SOC from degradation[24–29] (Fig. 1). These processes are examined in detail, including SOC mineralization in the presence of roots, microbial C-use efficiency, spatial distribution of C functional groups, and mineral protection of SOC, to quantify the potential of biochar to lift the SOC storage ceiling. We demonstrate the importance of fine-scale spatial heterogeneity and temporal variability of diverse C functional groups associated with mineral fractions for building and protecting rhizo-deposits over a decade.

## Results and discussion
### Lifting the storage ceiling of soil organic carbon
To examine the potential for biochar to protect soil organic matter from microbial degradation, we measured SOC stocks in a

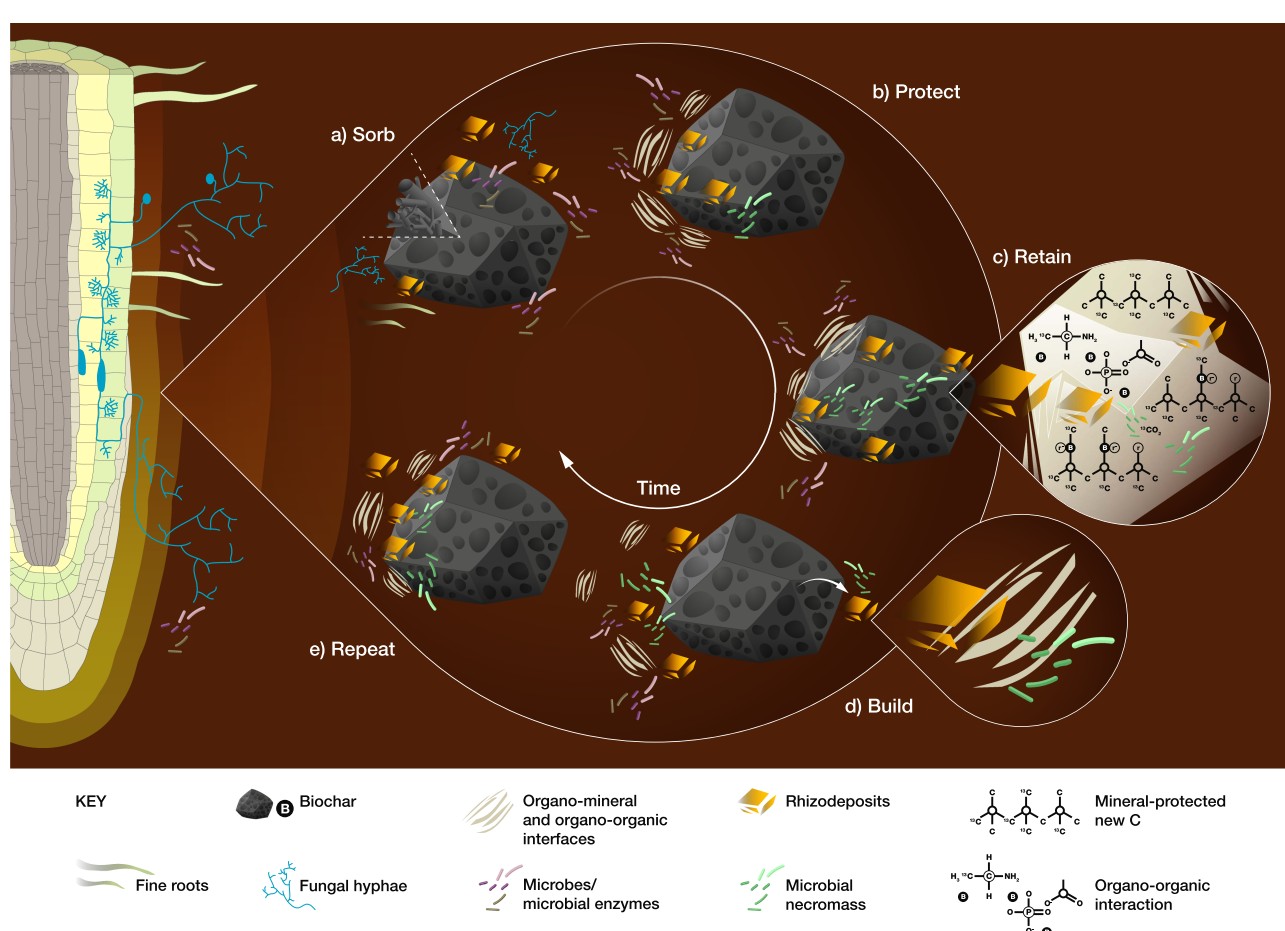

**Fig. 1 | Conceptual diagram of the formation of protected aggregates from catalytic biochar surfaces over time in a Rhodic Ferralsol. a** Biochar sorbs root-derived carbon (rhizodeposits) onto its surface, protecting the rhizodeposits from immediate microbial consumption. **b, c** The rhizodeposits form organic interfaces with biochar, and organo-mineral interfaces with very fine layers of soil minerals that accumulate on the biochar, that protect (**b**) and retain (**c**) rhizodeposits within the biochar coating. Over time, microbial necromass also adsorbs to biochar being retained in similar protective interfaces. **d** New organic and organo-mineral coatings can build on the biochar surface. **e** The process repeats, to develop new, protected SOC over time.

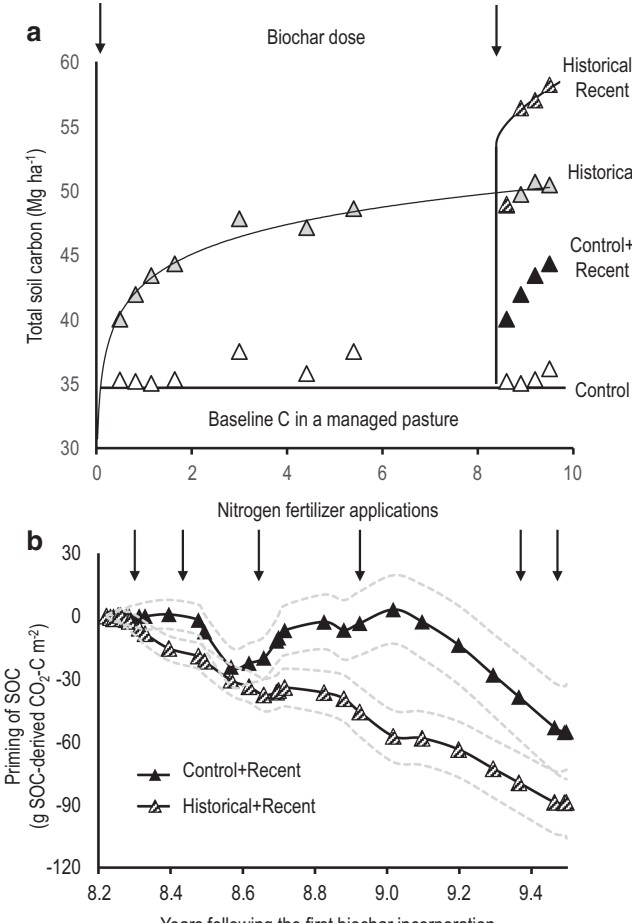

**Fig. 2 | Belowground carbon dynamics in a long-term continuous biochar field experiment. a** Changes in total soil organic carbon (SOC, Mg C ha⁻¹) in the 0–75 mm soil layer of the Control and biochar-amended soils over 9.5 y ($n = 3$, LSD = 1.1). **b** Rhizosphere priming, shown as the difference in cumulative SOC mineralization between planted and unplanted systems of Control+Recent and Historical+Recent soils. Shaded regions in **b** represent 95% confidence intervals normalized against the mean squares across both treatments at each sampling event ($n = 3$). Confidence intervals were based on a sensitivity analysis that considers the extreme scenarios of contrasting SOC pools (C₃ vs. C₄-dominated) by differences in δ¹³C soil signatures.

biochar-amended, managed pasture over 9.5 y. The field site, converted to managed pasture from subtropical forest 100 y ago, had a SOC stock 17% lower than the adjacent native rainforest in the 0–75 mm topsoil layer. Soil was subjected to four treatments as Control: no biochar application; Historical: biochar applied once at trial establishment (2006); Control+Recent: biochar applied after 8.2 y to the original Control plots; and Historical+Recent: biochar applied after 8.2 y to original Historical plots, i.e., two biochar applications (Supplementary Table 1). Otherwise, field plots were managed (sown with annual ryegrass, fertilized, and harvested) identically.

Our results showed that the SOC storage ceiling could be lifted through either single or multiple applications of biochar (Fig. 2a). The Control stored 35 (±1.3) Mg C ha⁻¹ in the topsoil (0–75 mm), while the Historical plots stored 50 (± 1.1) Mg C ha⁻¹ at 9.5 y after biochar addition (Fig. 2a). When biochar was added to the Control plots after 8.2 y (Control + Recent), the SOC storage capacity was raised to 44 (±0.7) Mg C ha⁻¹ 1.3 y following biochar application. A second application of biochar after 8.2 y (Historical + Recent) raised the total SOC to 58 (± 0.2) Mg C ha⁻¹ 1.3 y later. The total increase of 15 Mg C ha⁻¹ after 9.5 y in Historical plots consisted of 5.7 Mg biochar-C ha⁻¹ in the topsoil plus

an additional 9.3 Mg C ha⁻¹ from the enhanced SOC accumulation. Furthermore, this enhanced SOC accumulation could be increased by multiple applications of biochar–the total increase of 23 Mg C ha⁻¹ in the Historical+Recent treatment after two biochar applications over 9.5 y consisted of 11.4 Mg biochar-C ha⁻¹ and 11.6 Mg C ha⁻¹ from enhanced SOC accumulation (1.01 Mg new SOC per Mg biochar-C). Thus, the second application of biochar in the Historical+Recent plots increased the SOC storage capacity by an additional 2.3 Mg new C ha⁻¹ compared to the Historical soil with a single application of biochar, with this being a 25% increase in new SOC accumulation caused by the second application of biochar.

The increase in SOC storage was due to a decrease in net cumulative SOC mineralization (defined as negative priming). The Historical+Recent treatment lowered SOC mineralization in the presence of roots by 89 g $CO_2$-C m⁻² over 1.3 y compared with Control+Recent soils, in which SOC mineralization dropped by 55 g $CO_2$-C m⁻² ($P < 0.05$, Fig. 2b). Neither Historical+Recent nor Control+Recent soils exhibited changes in soil, soil+root, or root respiration compared to the Control soils ($P > 0.05$, Supplementary Figs. 2 and 3). As a portion of the total $CO_2$ flux, root respiration remained relatively consistent (27–36%) and was unaffected by treatments within each pulse labelling event ($P > 0.05$, Supplementary Table 2).

To further examine this decrease in SOC mineralization, we partitioned rhizodeposits (root C) from biochar C and SOC within aggregate size and density fractions. Historical+Recent soils had a similar proportion of total recovered ¹³C (58 ± 5.7 %, Fig. 3a) as Historical soils (60 ± 9.8%), with this being around 18% greater than Control (42 ± 7.3%) and Control+Recent soils (45 ± 4.5%; Fig. 3b) after the pulse-labelling event at 9.5 y ($P < 0.05$; Supplementary Table 3). This increase in belowground ¹³C retention could be largely explained by an increase in ¹³C associated with mineral-protected soil organic matter (M-SOM), which increased by 14% in Historical+Recent compared with Control+Recent soils ($P < 0.05$, Supplementary Table 4). Initially, Historical+Recent soils nearly doubled the ¹³C retention in the occluded particulate organic matter (O-POM) fractions of microaggregates (5 mg ¹³C m⁻²) and M-SOM fractions of macroaggregates (14 mg ¹³C m⁻²) at 8.9 y compared to Control+Recent soils (Supplementary Fig. 4a, b). The root-derived ¹³C from rhizodeposition was gradually accumulated into O-POM and M-SOM in macroaggregates at 9.2 y (Supplementary Fig. 4c, d), which transformed into the M-SOM fraction in micro- and macroaggregates by 9.5 y (Supplementary Fig 4e, f).

## Microbial contribution and responses to the retention of rhizodeposits

To determine the microbial contribution to this increased SOC storage capacity, we quantified catabolic enzyme activities, metabolic quotients of native SOC (bulk soil) and rhizodeposition (¹³C content), and specific enzyme activity in Control+Recent and Historical+Recent soils. Microbial biomass increased by 8–12% in Control+Recent compared with Historical+Recent soils between 8.9 and 9.5 y (Supplementary Table 5), as a result of the stimulation of microbial co-metabolism[30] by the addition of biochar-C to a previously unamended soil, which also induced a small positive priming effect in Control+Recent soils (Fig. 2b). Historical+Recent soils increased substrate-induced respiration for citric, malic, and protocatechuic acids compared to Control+Recent soils, but no differences were detected for 12 other C substrates that are all common in agricultural soils (Supplementary Fig. 5). This greater respiration induced by carboxylic and phenolic acids (common in root exudates) partially explained the higher metabolic quotient associated with bulk SOC in Control+Recent than Historical+Recent soils (Supplementary Table 5). Lower metabolic quotients indicate higher substrate-use efficiency, so the lower metabolic quotient observed in Historical+Recent soils supports the more rapid establishment of negative priming than in the Control

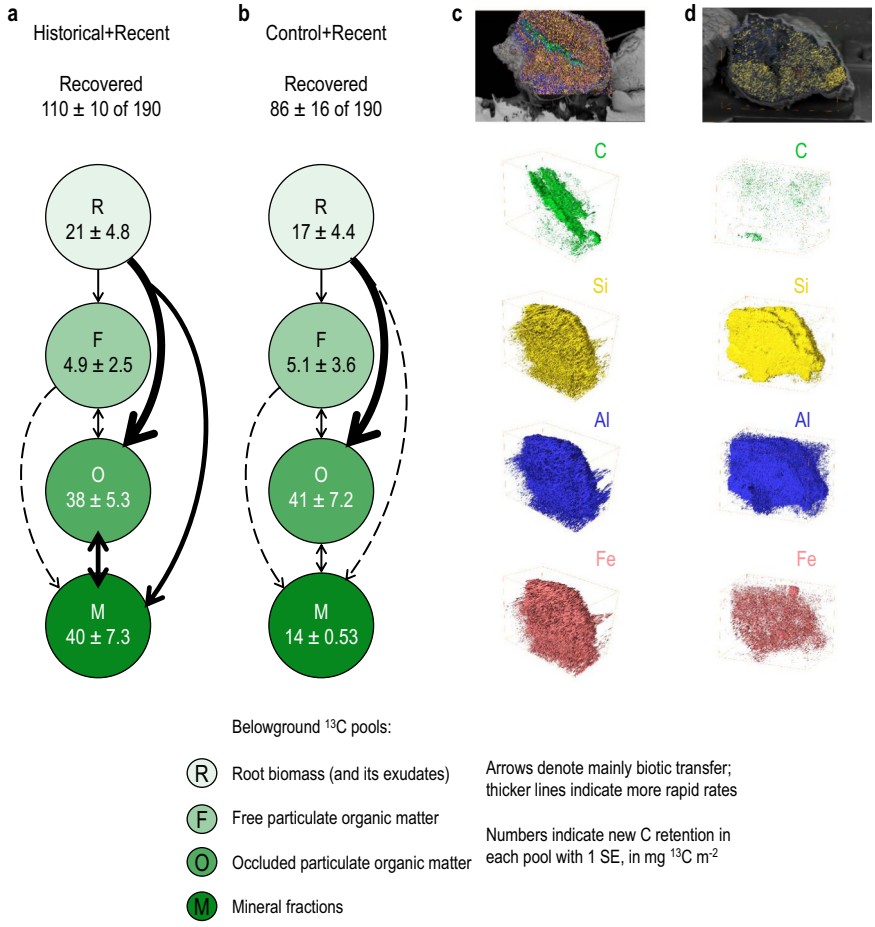

**Fig. 3 | Allocation and retention of rhizodeposits ($^{13}$C-enriched) and three-dimensional (3D) elemental distribution in a biochar-amended Ferralsol at 9.5 y.** New C retention in different belowground carbon pools from Historical +Recent (**a**) and Control+Recent (**b**) soil ($n = 3$). The total recovery of $^{13}$C from $^{13}$C-CO$_2$ pulse labelling (± SE) is given for soil + root respiration, root biomass, free and occluded particulate matter, and mineral fractions 15 days after labelling each plot with 190 mg $^{13}$C-CO$_2$ m$^{-2}$. **c, d** Three-dimensional focused ion beam scanning electron microscopy energy dispersive spectroscopy (3D FIB-SEM-EDS) of an intact soil aggregate from a Historical+Recent biochar plot (**c**, 30 μm × 25 μm × 24 μm) or Control+Recent plot (**d**, 30 μm × 20 μm × 30 μm), showing distribution of carbon, silicon, aluminium, and iron.

+Recent soils 8.2 y after the biochar addition (Fig. 2b). This suggests that the microbial accessibility to SOC might be limited in the Historical+Recent plots, whereas in the Control+Recent biochar, the soil microorganisms had to adapt to a change in C-substrate type and availability[31].

The ratio of extracellular enzymes to microbial biomass (specific enzyme activity) can be used to indicate the C-turnover efficiency of the soil microbial community, and a low specific enzyme activity can retard the mineralization of native SOC[32]. Here, the ratio of enzyme activity-to-microbial biomass was similar in both Historical+Recent and Control+Recent soils compared to the Control soil (Supplementary Table 6) despite the reduced enzyme activities (Supplementary Table 7). This is consistent with decreased metabolic quotients/ increased microbial C-use efficiency (Supplementary Table 5) for bulk SOC but not root-derived C in the amended soils, which indirectly contributes to negative priming (Fig. 2b). The presence of opportunistic microbes that meet their energy and nutrient demands by exploiting the catalytic activities of decomposers could lower the specific enzyme activity[32]. It is noted that sorption affinities of the fluorophore and/or the enzyme to biochar compared to other soil surfaces may lead to underestimating enzyme activities[33]. Here, we used matrix-matched standard curves to account for any potential binding (or quenching/excitation) of the fluorophore. The fluorescence response of standard curves constructed using the soil matrix with or without biochar were not significantly different (Supplementary Table 9), suggesting that fluorophore sorption, quenching, or excitation did not contribute to the observed differences in enzyme activities.

## Spatial heterogeneity of SOC

Our study provides the first visual evidence of a mechanism by which biochar can accelerate the formation of organo-mineral and organic interfaces in soils to protect SOC from microbial degradation, summarized in Fig. 1. Biochar can sorb root-derived C (rhizodeposits) that forms biofilms on its surfaces (Figs. 1a and 3a). The very fine layer of soil minerals that accumulate on biochar as it ages in soil[30,34,35] protects rhizodeposits from microbial metabolism[36,37] over time. Microbial necromass is also incorporated into this coating of organo-mineral and organic interfaces and is protected from degradation[38–44] (Figs. 1b, c, 4, and 5). A coating can build on the biochar surfaces (Fig. 1d) and the processes repeat to build rhizodeposits in soil over time (Fig. 1e). Our spectroscopic data showed the formation of clay–organic complexes as one possible mechanism by which biochar promotes the accumulation of new biogenic SOC.

To visualize the retention of rhizodeposits and microbial-derived C, we undertook one-dimensional (1D) spectroscopic, two-dimensional (2D) microspectroscopic, and three-dimensional (3D) electron microscopic analyses of SOC spatial heterogeneity. We provided direct visual evidence of the spatial heterogeneity at the nano- to micro-scales. To better understand the process of negative priming

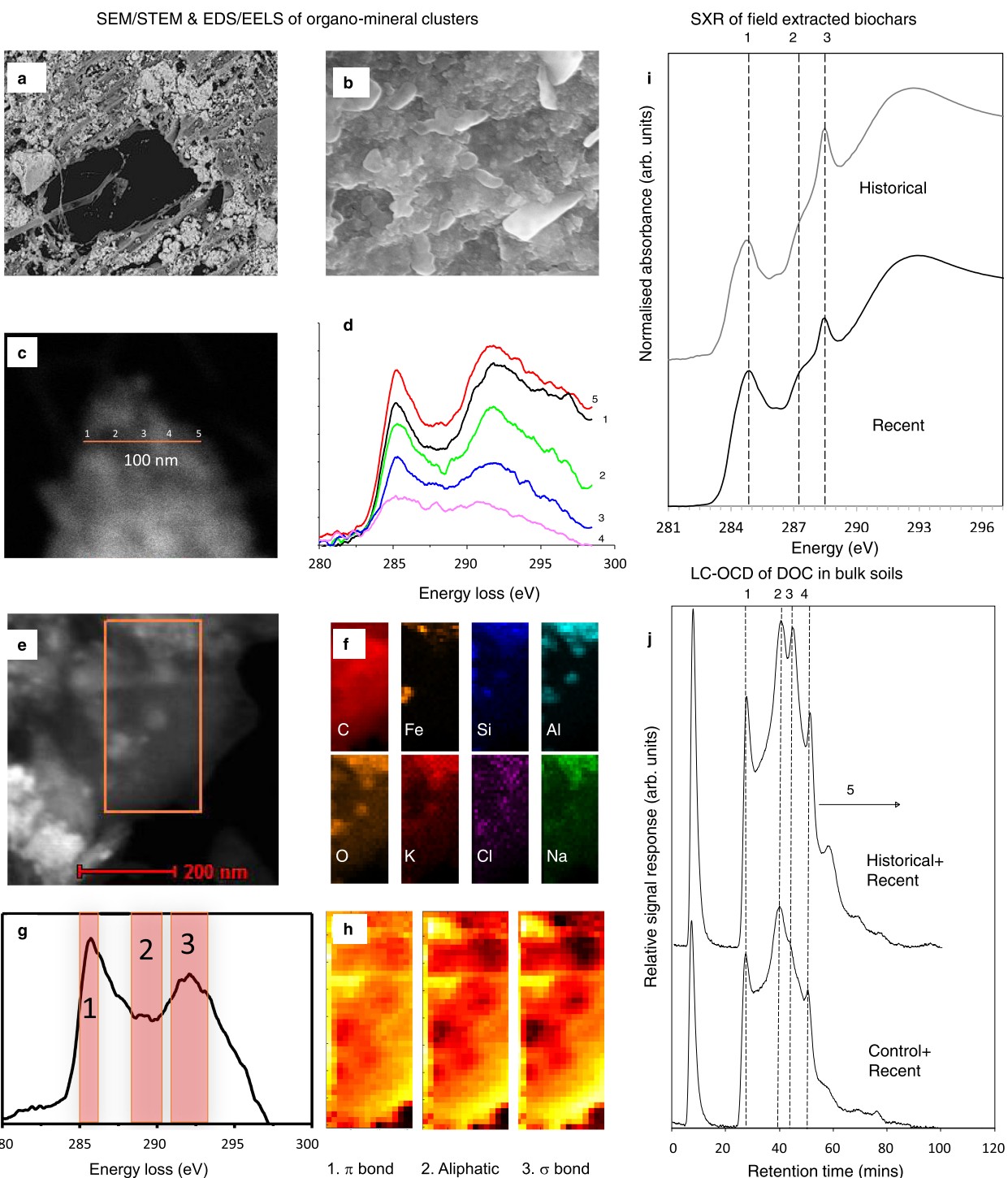

**Fig. 4 | In situ spectromicroscopic analysis of the organo-mineral coating on biochar surfaces and pores. a** A pore in biochar that was recovered from the Historical plots (scanning electron microscopy [SEM]). Bar, 50 μm. **b** Surface of the organo-mineral layer inside the biochar pore from Historical plots (SEM). Bar, 200 nm. **c** Organo-mineral clusters on a biochar surface from the Control+Recent plots (high-angle annular dark-field scanning transmission electron microscopy [HAADF-STEM]); **d** the electron energy loss spectra (EELS) at positions 1–5 of the clusters in (**c**). **e** A deposit attached to the surface of biochar from the Historical plots (HAADF). **f** 2D elemental mapping of the boxed area in **e** (energy-dispersive spectroscopy [EDS]. **g** EELS of the boxed area in (**e**). **h** Mapping integration of EELS regions 1–3 in (**g**). **i** Average soft X-ray (SXR) emission spectra of field-extracted Control+Recent (1-y) and Historical (9.5-y) biochars (*n* = 9, CV < 3%). **j** Dissolved organic content (DOC) of Historical+Recent and Control+Recent soils (liquid chromatography-organic carbon detection [LC-OCD]). The hydrophilic fraction is further sub-divided into five categories: biopolymers, persistent C, building blocks, low molecular weight acids, and low molecular weight neutral molecules.

following biochar application, we mapped the elemental composition within intact aggregates to determine whether the retention of rhizo-deposits (and other forms of C) may be facilitated via protection by Fe and Al-rich soil minerals. The 3D distribution of C, Si, Al, and Fe was assessed using a focused ion beam (FIB) coupled with scanning electron microscopy (SEM) and elemental detection provided by energy-dispersive X-ray spectroscopy (FIB-SEM-EDS; Fig. 3c, d). These results illustrate how C (including rhizodeposits) can be retained through the

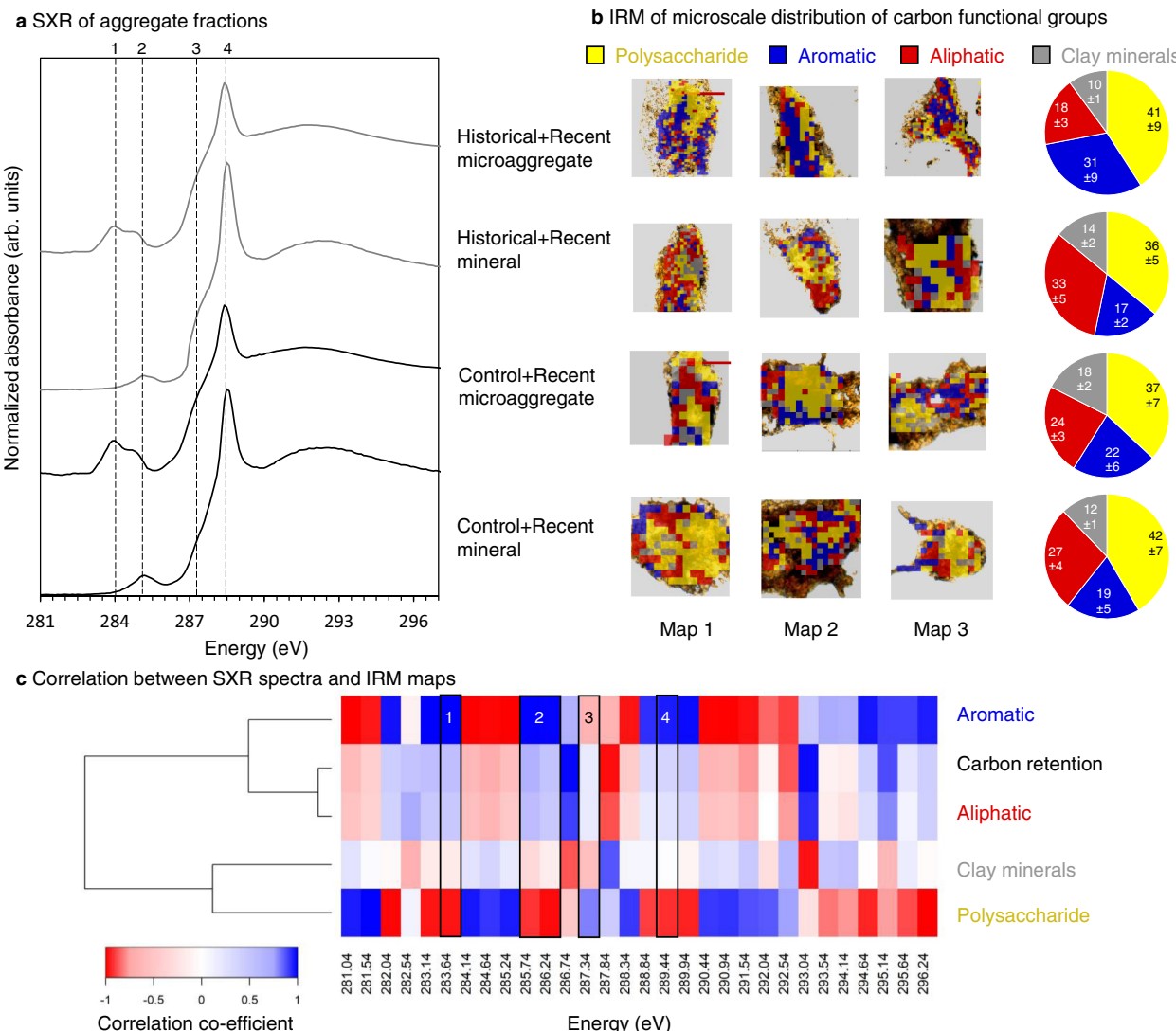

**Fig. 5 | Synchrotron-based spectromicroscopic analysis of microaggregates (53–250 μm) and mineral fractions (<53 μm) in Control+Recent and Historical +Recent soils. a** Average soft X-ray (SXR) spectra of microaggregates and mineral fractions from Historical+Recent and Control+Recent plots (*n* = 9, CV < 3%), featuring quinones, aromatic C, aliphatic C, and carboxylic C. **b** Semi-thin (200 nm) sections of free water-stable microaggregates and mineral fractions isolated from Historical+Recent and Control+Recent plots analyzed using synchrotron-based infrared-microspectroscopy (IRM). Spectral maps showing the distribution of polysaccharide-C (1035 cm⁻¹), aromatic-C (1600 cm⁻¹), aliphatic-C (2920 cm⁻¹), and clay mineral-OH (3650 cm⁻¹) obtained from 64 co-added scans, overlaid on optical micrographs of the semi-thin sections. Bars, 50 μm. Pie charts display normalized optical proportions of the four features analyzed by an image processing pipeline (Supplementary Fig. 10 and Supplementary Table 12). **c** Hierarchical clustering dendrogram indicating similarity relationships between C functional groups from SXR (first derivatives), and their distribution from IRM or belowground C retention (Fig. 3a, b) across biochar treatments.

formation of organo-mineral complexes with Fe oxides in the soil (Fig. 1b; Supplementary Movie 1).

We further examined biochar extracted from the Control+Recent soil on a nanometre scale by directly analyzing the chemical composition of the organo-mineral and organo-organic coatings on the biochar surface and in its pores. An image of the area where fungi were located inside a biochar pore shows a high concentration of irregular pores and a coating of organic material (Fig. 4a, b). Fungi can mine nutrients from minerals by exuding acids[45,46] that may cause the observed microporosity of organo–mineral–biochar interfaces (Figs. 4a–c, e, S6). Complex changes had occurred on the biochar surface over 1.3 y, revealed by EDS analysis (Figs. 4f, S6c–f). One possible mechanism is that positively-charged nanoparticulate minerals rich in Al, Si, Ca, P, and Fe can be attracted to the surface of the negatively-charged biochar. These positively-charged minerals subsequently attracted negatively-charged organic molecules with detectable concentrations of C=C, C–OH, C–N/C=N, C=O, COOH functional

groups, quinone bonds, and anions, thus initiating a process whereby porous clusters were accumulated on the biochar surface (Fig. 4d, g, h; Supplementary Table 9). It is also noted that Fe-Al-oxyhydroxide minerals can have both positive and negative net surface charge depending on pH (i.e., they are variable charge minerals). Most biochars are dominated by neutral carboxyl or negative carboxylate groups (depending on pH), but some biochars may also have positively-charged oxonium groups. Using synchrotron-based soft X-ray (SXR) spectroscopy (Fig. 4i), we observed greater intensities of carboxyl COOH (288.6 eV) in the 9.5-y aged biochar (10.6%) compared with the 1.3-y aged biochar (6.1%; Supplementary Table 11). Similarly, exudates from plants and microorganisms can be deposited around minerals and attracted by cations onto biochar surfaces. Recent biochar application to the Historical plots would provide new unoccupied surfaces and pores in the soil to increase sorption capacity for root exudates[47] (Fig. 1a), which would then serve as binding agents to further enhance aggregation[48]. As these clusters build, they may also be

detached from the biochar either through fluctuating redox conditions, interaction with microbes, or perturbation caused by soil invertebrates or human activities[34] (Fig. 1d). These results provide direct evidence of repeated cycles of formation of organo-mineral coatings on the biochar surfaces during aggregate turnover or in response to changes in soil conditions, with these processes accumulating rhizodeposits in soil over time.

These biochar micro-sites have a high concentration of free radicals with low-molecular-weight organic C and/or inorganics dissolved from the biochar (Supplementary Table 9). Colloidal biochar particles, leachates, dissolved native SOC, and rhizodeposits may be further retained separately or held together via cation bridging with $Ca^{2+}$ or Al and Fe oxyhydroxides[49–51] and/or organic interactions at the nanoscale[52] (Fig. 4f, h). These processes may be encouraged by oxidation of the biochar surface as it ages in soil[39,40]. This hypothesis is supported by liquid chromatography-organic carbon detection (LC-OCD), which reveals that total dissolved organic C, hydrophobic C fractions, and building blocks (oxidized persistent C including polyaromatic acids and polyphenols) were higher in Historical+Recent than in Control+Recent soils (Fig. 4j; Supplementary Table 10). The analysis of the surface of the 9.5-y aged biochar by C-edge energy electron loss spectroscopy (EELS) and X-ray photoelectron spectroscopy (XPS) indicated that most of the oxidized C species were formed in the organo-mineral coating (Fig. 4f–h). The concentrations of the different functional groups were influenced by the presence of nanophase Fe, Si, and Al oxides[52].

We further validated the nanoscale observations of biochar surfaces in soil at the microscale. To differentiate the molecular diversity of organic compounds and their lateral arrangement with respect to organo-mineral interfaces, we conducted in situ spectromicroscopic analysis using SXR (Fig. 5a) and synchrotron-based infrared microspectroscopy (IRM; Figs. 5b, S7, S8). We examined intact water-stable microaggregates (53–250 μm) and the mineral fraction (<53 μm) from Historical+Recent and Control+Recent soils. SXR analyses revealed that the C functional groups in the microaggregates were dominated by quinones, aromatic C (1s-$\pi^*$ transitions of conjugated C=C), aliphatic C, and carboxylic C (Fig. 5a), which are derived from biochar and rhizodeposits. The relative proportion of functional groups was similar between Control+Recent and Historical+Recent soils (Supplementary Table 11). The mineral fraction was characterized by dominant peaks of aliphatic, amide, and carboxylic C, suggesting deposition of microbial metabolites or debris, exopolysaccharides, and root exudates onto mineral surfaces[27,53–57]. The proportion of aliphatic C nearly doubled in the Historical+Recent treatment (30.6%) compared with the Control+Recent treatment (15.6%; Supplementary Table 11). These data provide evidence of rhizodeposits and microbial necromass incorporation into SOC, with rhizodeposits predominantly in microaggregates rather than mineral fractions. This difference indicates that retention of rhizodeposits in SOC relies on forming complex organic and organo-mineral interfaces with microbial necromass and biochar, while microbial necromass can be protected by organo-mineral interfaces in mineral fractions.

These SXR results also align with the micro-spatial maps produced from IRM analyses of sections taken from intact aggregates (Fig. 5c). The correlation between clay minerals and microbial metabolites (aliphatic-C) in the microaggregates was stronger in Historical+Recent soils than in Control+Recent microaggregates ($R^2 = 0.96$ vs. 0.86, Supplementary Fig. 9). In contrast, the correlation of microbial-derived C with clay was similar for both mineral fractions ($R^2 = 0.94$–0.95). The correlation between polysaccharide-C and clay was much greater in Historical+Recent than in Control+Recent microaggregates ($R^2 = 0.83$ vs. 0.46, Supplementary Fig. 9). We developed an image processing pipeline to quantify the distribution of C forms in association with clay from IRM (Supplementary Fig. 10). For the microaggregates, a greater proportion of aromatic-C (31% of pixels across the intact section) was found in Historical+Recent soil compared with Control+Recent soil (22%) because of the biochar persisting in Historical soils from the original soil amendment at the trial establishment (Fig. 5b; Supplementary Table 12). The distribution of polysaccharide-C (36–42%), aromatic-C (17–19%), aliphatic C (27–33%), and clay (12–14%) was similar in the two mineral fractions (Fig. 5b; Supplementary Table 12). These observations agree with new C retention in belowground [13]C pools (Figs. 3a, b and 5c), highlighting the importance of clay minerals for protecting SOC from microbial mineralization.

## Potential global impact of lifting the soil carbon ceiling

The elevation of the SOC ceiling observed in our trial has significant implications for the global efforts to build SOC[9,58,59]. We have estimated the magnitude of the potential CDR that could be delivered if the SOC increase demonstrated here is extrapolated to similar contexts globally. Based on global potential production of woody feedstock of 0.48–0.90 Pg C $y^{-1}$, assuming that this biochar is applied to Ferralsols under tropical pasture with the same response of about 1.01 Mg new SOC per Mg biochar-C over two applications applied, an additional soil C sink of 0.23–0.45 Pg C $y^{-1}$ could be potentially achieved. This increase represents a substantial increase over the current contribution of grasslands of 0.04 Pg C to the global SOC pool[6].

In our study, we raised the SOC storage capacity in a subtropical soil with a strategic application of a *Eucalyptus saligna* biochar (550 °C) 8.2 y after the original biochar application. Of importance to building soil C stocks, a second application of biochar to previously amended soils resulted in 2.3 Mg new C $ha^{-1}$ (i.e., microbial necromass and rhizodeposits) being stored as SOC (Historical + Recent vs. Historical, Fig. 2a). Our in situ spectromicroscopic analyses at the molecular to microaggregate scales showed accumulation of clay mineral-organic complexes in the soil. This spectroscopic evidence supports our proposed model (Fig. 1) for one possible mechanism by which biochar promotes the accumulation of new biogenic SOC. This mechanism, if found to apply in other tropical Ferralsols, could substantially increase the potential for CDR through the use of biochar.

## Methods
### Field site details
The field experiment was situated at the Wollongbar Primary Industries Institute (28°49′S, 153°23′E, elevation: 140 m), Wollongbar, New South Wales, Australia. The classification and properties of the soil can be found in Weng et al[60].. Briefly, the soil is a Rhodic Ferralsol, a fine-textured and Fe-rich mineral soil dominated by kaolinite, gibbsite, and goethite. The 100 mm topsoil was $pH_{CaCl2}$ 4.5, with 35 g $kg^{-1}$ C, 84 g $kg^{-1}$ Fe, and 67 g $kg^{-1}$ Al.

Details of the initial field site establishment in 2006 are in Slavich et al[61]. Each of the three replicate plots was treated either with biochar incorporated into the topsoil (0–100 mm) at 10 Mg $ha^{-1}$ (1% w/w, 7.6 Mg biochar-C $ha^{-1}$, applied to 100 mm depth) plus nitrogen phosphorus potassium (NPK) fertilizer ('Historical'), or NPK only ('Control'). An annual ryegrass (*Lolium multiflorum*) was seeded each year at 35 kg seed $ha^{-1}$. Urea was applied at 46 kg N $ha^{-1}$ six times each year (276 kg $ha^{-1}$ total) in winter and spring following manual cutting of pasture grass to simulate grazing. Basal nutrients containing P and K were applied annually at sowing[61]. At 8.2 y after trial establishment (April 2014), each Historical and Control plot was superimposed with subplots (0.5 m × 0.5 m), and biochar added again at the same rate to subplots ('Historical + Recent' and 'Control + Recent'). Field sites were maintained as previously described for a further 1.3 y.

The same biochar batch was added to the field site in 2006 and 2014. Biochar was derived from a single source of aboveground biomass of mature *Eucalyptus saligna*, pyrolyzed at 550 °C for 30 min (Pacific Pyrolysis, NSW, Australia), and sieved to <2 mm before

application. The biochar density was 0.332 g cm$^{-3}$ (following Quin et al.[62]), and its chemical properties are described in Slavich et al.[61]. For storage, biochar was air-dried and archived in sealed 200 L steel containers at room temperature. Biochar (10 Mg ha$^{-1}$) was mixed with 100 mm topsoil and repacked into plots to a bulk density of 1 g cm$^{-3}$; plots to which no biochar was added were also excavated and repacked to the same bulk density. The topsoil±biochar was weighed before repacking to determine soil bulk density for each treatment.

## Soil and root respiration collars
Specialized respiration collars were used to isolate soil-only and soil+root respiration from shoot respiration[43,60] (Supplementary Fig. 1). A sand+root collar (50 mm diameter) packed with acid-washed sand and planted with ryegrass was installed in each Control subplot to measure root-only respiration. A similar sand+root collar was packed with a biochar–sand mixture (1% w/w) in each Historical+Recent subplot. NPK fertilizers were applied as described above to maintain root growth into the collars. Moisture content was maintained at 60–80% field capacity in the root collars to minimize C isotopic fractionation during photosynthesis caused by water stress[63].

## Soil sampling
Soils were sampled at 8.9, 9.2, and 9.5 y after trial establishment. Intact soil cores (40 mm diameter) were sampled to 75 mm depth within each subplot, outside the respiration collars to reduce disturbance. Note that although 7.6 Mg biochar-C ha$^{-1}$ was incorporated to 100 mm depth, soils were sampled to 75 mm depth because the trial originally started as an 'agronomic assessment of biochar' and the industry standard for pasture soil analysis was 0–75 mm sampling. Hence, the amount of biochar-C in the top 75 mm layer was estimated to be 5.7 Mg biochar-C ha$^{-1}$ assuming no lateral movement of biochar. This may underestimate new SOC accumulation. Previously sampled areas were avoided in subsequent sampling events. Samples were mixed evenly and analyzed for pH, total SOC, and microbial biomass C (MBC). Total SOC was measured on an equivalent-mass basis using Dumas combustion[60], and converted to soil C density using the bulk density of each biochar treatment. Soil pH was measured on soil suspensions (1:5 w/w soil:water) using an IntelliCAL PHC101 pH probe on a Hach HQ40d portable metre (Loveland, Colorado, USA). The metabolic quotients of total C or rhizodeposits were then quantified as the ratio of respiration (native SOC or $^{13}$C-labelled root respiration) over total MBC. The remaining soil was stored at −20 °C.

## SOC priming in the plant–biochar–soil systems
To understand how plant–biochar–soil interactions affect SOC priming, the $\delta^{13}$C signature of $CO_2$-C from soil-only, soil+root, and sand+root samples was measured before and after pulse labelling events (Supplementary Fig. 1). The C content and $\delta^{13}$C signatures of bulk soil, aggregates, and fractions, were measured using a PDZ Europa ANCA-GSL elemental analyzer interfaced to a PDZ Europa 20-20 isotope ratio mass spectrometer (Sercon Ltd., Cheshire, UK), according to Weng et al.[60].

Three pulse labelling campaigns were conducted on three occasions: 12 June 2014, 01 August 2014, and 30 July 2015. Each event applied 190 mg $^{13}$C m$^{-2}$ as the label and was analyzed as an independent experiment assuming no retention of $^{13}$C from prior events. The excess of enriched $^{13}$C-$CO_2$ from soil+root ($\delta^{13}$C$_{Total}$) and sand+root ($\delta^{13}$C$_{Biochar+Root}$) respiration was measured 3, 5, 10, and 15 d after each pulse labelling event. Soil-only respiration was measured in Control plots with no pulse labelling ($\delta^{13}$C$_{Soil}$) and at 15 d after pulse labelling ($\delta^{13}$C$_{Total'}$). Biochars were recovered by hand from soil samples, thoroughly rinsed with distilled water on a 100 μm sieve, and oven-dried at 50 °C for 24 h. The $\delta^{13}$C signatures of aged biochar (from Historical subplots) and new biochar (from Control+Recent subplots) were both −25.0 ± 0.1‰.

The rhizosphere priming of native SOC was quantified using a three-pool C partitioning model: biochar-C, root-C, and SOC. Any interactive effect of biochar and root on the $\delta^{13}$C signature of soil would be surpassed by a greater level of $\delta^{13}$C enrichment of the root component compared with any isotopic signature contribution from soil and biochar to the $\delta^{13}$C signature of the total respiration.

The mineralization of native SOC (C$_{Soil}$) in the presence of plant roots was calculated by:

$$C_{Soil}(\%) = 100 \times (\delta^{13}C_{Total} - \delta^{13}C_{Biochar+Root})/(\delta^{13}C_{Soil} - \delta^{13}C_{Biochar+Root})$$
(1)

Similarly, the proportion of soil-derived $CO_2$-C in total respiration from plant-free soil (C$_{Soil'}$) was determined by:

$$C_{Soil'}(\%) = 100 \times (\delta^{13}C_{Total'} - \delta^{13}C_{Biochar})/(\delta^{13}C_{Soil} - \delta^{13}C_{Biochar})$$
(2)

Rhizosphere priming of SOC in the biochar system was the difference in native SOC mineralization between the plant-containing and plant-lacking systems, partitioned from biochar endmembers:

$$Priming = (C_{Soil}(\%) \times C_{Total} - C_{Soil'}(\%) \times C_{Total'})/100$$
(3)

## Sensitivity analysis of isotopic partitioning
A sensitivity analysis of C source partitioning was performed to assess the impact of plant–biochar (C$_3$-dominated)–soil interactions on $\delta^{13}$C signatures of soil (a mixture of C$_3$ and C$_4$ pools). Errors generated from isotopic partitioning were propagated using the first-order Tyler series approximations of the variances of native SOC mineralization.

Because of the uncertainty of the direction of biochar-induced priming of soil C and/or rhizodeposits, the contribution of biochar on the $^{13}$C endmember of $\delta^{13}$C$_{Soil}$ was assessed. Three alternative scenarios of three-pool C partitioning were evaluated:

(1) Dominant positive priming of new C from the C$_3$ pasture, where $\delta^{13}$C$_{Soil'}$ = −27‰ (i.e., the upper bound of the 95% confidence interval, Fig. 2b);

(2) Equal native SOC priming and rhizosphere priming, hence, the same $^{13}$C signatures of soil+root in the biochar-amended and Control plots, where $\delta^{13}$C$_{Soil'}$ = $\delta^{13}$C$_{Soil}$ (i.e., solid lines in Fig. 2b);

(3) Dominant positive priming of the native C$_4$-dominant SOC, where $\delta^{13}$C$_{Soil+Root'}$ = −13‰ (i.e., the lower bound of the 95% confidence interval, Fig. 2b).

The boundary conditions were calculated from published $^{13}$C signatures[63] for Scenarios 1 and 3, and the 95% confidence intervals are the combination of the lowest and highest scenarios (n = 3). First-order Tyler series of the variances ($\sigma^2$) of the proportion of soil respiration, C$_{Soil}$(%), were approximated to propagate errors from isotopic partitioning[64].

$$\sigma^2 C_{Soil}(\%) = (\sigma^2 \delta^{13}C_{Total} - \sigma^2 \delta^{13}C_{Soil})/(\delta^{13}C_{Total} - \delta^{13}C_{Soil})^2$$
(4)

## Enzyme activity and substrate-induced respiration
Six samples were derived from the Control, Historical+Recent, and Control+Recent subplots at 9.5 y in both the soil-only and soil+root collars. The determination of catabolic enzyme activities using a soil suspension method is described in Weng et al.[43]. Briefly, after 7-d incubation at 40% water-holding capacity, the activities of four hydrolytic (β-glucosidase, xylosidase, cellulase, and N-acetyl-glucosaminidase) and phosphatase enzymes in soils were quantified using a fluorogenic substrate (4-methylumbelliferyl [MUF]). Standard curves were used to determine enzyme activity on a microplate reader (BMG labtech FLUOstar Omega) in the presence and absence of soil suspension. The MBC was analyzed using a chloroform fumigation

method. Briefly, 20 g of fresh soil (dry-weight equivalent), was fumigated within four days after collection with alcohol-free chloroform in a desiccator for 24 h in the dark at 22 °C. The same amount of non-fumigated soil and the fumigated soil were shaken for one hour with 80 ml of 0.5 M $K_2SO_4$ solution. The extracts were filtered through a glass-fibre filter paper (Whatman GF/C) and stored at −18 °C until analysis. Specific enzyme activity was obtained by dividing the activity of the individual enzyme over the total MBC at each sampling time.

Substrate-induced respiration was used to measure community-level physiological profiles using the MicroResp™ method[65] with minor modifications. Fresh soil samples ($n = 8$ for each soil type), packed in 96-deepwell plates (0.5 g per well), were prepared as for enzyme activity analysis[43]. Fifteen C substrates (Supplementary Table 6) were selected to represent a broad range of soil and root exudates that comprise a large proportion of SOC[65]. Hydrolysis respiration was determined with a distilled water blank and was subtracted from values for C substrates.

### Aggregate size and density fractionation

Aggregate size (dry sieving) and density fractionation were conducted based on Weng et al.[42]. No large macroaggregates (>2000 μm) were found in this study. Macroaggregates (250–2000 μm) and microaggregates (53–250 μm) were fractioned into free particulate organic matter (F-POM; $\rho < 1.6$ kg m$^{-3}$), occluded POM (O-POM; >53 μm, $\rho > 1.6$ kg m$^{-3}$), and mineral-protected soil organic matter (M-SOM, combining silt- and clay-bound SOM; <53 μm, $\rho > 1.6$ kg m$^{-3}$).

### Analysis of belowground $^{13}$C pools

The recovery of $^{13}$C in various SOC pools at time $t$ ($A^{13}C_{i,t}$, in %) was calculated by dividing the amount of $^{13}$C (g m$^{-2}$) in a specific C pool ($C_i$) by the initial amount of total added $^{13}CO_2$ (g m$^{-2}$) at each labelling event ($^{13}C_{added}$):

$$A^{13}C_{i,t} = (^{13}C_{excess,t} \times C_i)/ ^{13}C_{added} \times 100 \qquad (5)$$

where i represents soil aggregates or their associated fractions; and $^{13}C_{excess,\ t}$ represents the increment of the $^{13}$C atom % of an individual C pool from its natural abundance level at a sampling time $t$.

### Three-dimensional focused ion beam scanning electron microscopy energy dispersive spectroscopy (3D-FIB-SEM-EDS)

Soil particle sections for EDS mapping were prepared in an FEI SCIOS FIB/SEM DualBeam system, with a vertical mount SEM column and an ion column at 52° to the electron column. The particle was located with the aid of the electron beam. Before milling, a 1 μm-thick Pt layer was deposited on the sample surface covering the area of interest to prevent subsequent damage by ion bombardment, and to reduce the curtaining effect during milling. The milling of the volume was performed with a 3 nA, 30 kV ion beam current; and EDS mapping data were collected with using a 5 kV electron beam with a 6.4 nA beam current. The voxel size of the SEM images is 84 nm ($x$) × 84 nm ($y$) × 1000 nm ($z$, slicing thickness).

### Synchrotron soft X-ray analyses

SXR analysis was performed at the SXR Spectroscopy beamline (14ID) at the Australian Synchrotron on the microaggregate (53–250 μm) and mineral fractions (<53 μm) from Historical+Recent and Control+Recent soils. Biochars were recovered from Control+Recent and Historical soils. Composite of five samples were collected in each field replicate and three laboratory replicates were obtained for each of three field replicates ($n = 9$). Samples were ground to a fine powder and mounted on double-sided carbon tape affixed to a stainless-steel ruler.

SXR spectra were collected at an angle of 100° to the beam over a photon energy range of 275–325 eV with a step size of 0.1 eV. The energy was calibrated using a graphite standard in the beamline, which was collected simultaneously with the normalization channel ($I_0$) and sample SXR spectra. An electron flood gun was used to minimize surface charging. Double normalization and a pre- and post-edge linear subtraction (background) were conducted using the Athena software (Stöhr 2013). Deconvolution and peak fitting of the double normalized spectra was carried out using in-house script on Matlab. A non-linear least square fitting of multiple Gaussians and one arctangent function were used to fit all the SXR data, following the procedure described in Solomon et al.[66]. Measures of the goodness of fitting using $R^2$ errors of better than 0.999 were achieved for the data.

### Synchrotron infrared microspectroscopy

For IRM, approximately 30 intact water-stable microaggregates (53–250 μm) and mineral fractions (<53 μm) from Historical+Recent and Control+Recent soils were hand-picked on a glass fibre filter paper and humidified gently over 18 h[67]. Aggregates and fractions were frozen at −20 °C before being cryo-ultramicrotomed at 200 nm using a diamond knife. No embedding medium was used. Multiple sections ($n > 6$) per sample were directly collected on CaF$_2$ windows (IR-transparent).

Samples were analyzed in triplicate at the IRM beamline at the Australian Synchrotron using a Bruker Hyperion 3000 infrared microscope and a V80v Fourier transform infrared spectrometer[67]. Spectral maps were acquired in transmission mode using 64 co-added scans with a resolution of 4 cm$^{-1}$, a beam size of 5.6 μm, and a step size of 5 μm. Multiple maps were acquired for each treatment to capture sample heterogeneity.

The IRM analysis was conducted using triplicate soil samples (i.e., three maps per treatment), resulting in a total of 815 individual spectrum measurements for microaggregates in the Historical+Recent treatment, 2335 spectra for the mineral fraction in the Historical +Recent treatment, 1331 spectra for microaggregates in the Control +Recent treatment, and 1874 spectra for the mineral fraction in the Control+Recent treatment.

Maps were processed using OPUS 8.2 software (Bruker Optik GmbH, Germany), targeting the absorbances at 3630 cm$^{-1}$ (−OH groups of clays), 2920 cm$^{-1}$ (aliphatic-C), 1600 cm$^{-1}$ (aromatic-C), and 1035 cm$^{-1}$ (polysaccharide-C)[67]. Integrated area under the absorption peaks representing each C functional group was used to produce a false-colour 2D map for image processing. Integrated areas were also used in conjunction with linear regression to assess the correlation between clay content and selected C functional groups.

### Image processing

The scales of optical intensities for different organo-mineral compounds were normalized across the four channels (polysaccharide-C, aromatic-C, aliphatic-C, and clay-OH), and background pixels excluded using a histogram-based thresholding algorithm[68] in FIJI[69]. Relevant pixels were divided by the sum image to compute the different local proportions of each channel. High intensity regions were defined as those with signal greater than the mean value of the normalized proportion of each channel. The masked, segmented images were combined to derive information about individual channels and different combinations of channels (see Supplementary Table 12 and Supplementary Fig. 10).

### High-angle annular dark-field scanning transmission electron microscopy (HAADF-STEM)−EDS−energy electron loss spectroscopy (EELS)

Forty biochar particles were extracted from each treatment and examined using a Zeiss Sigma SEM. Detailed analysis of five particles

was carried out using a Bruker X-ray Dispersive analyzer (EDS). A Cs-corrected FEI Titan 80/300 STEM, working at 80 keV and equipped with a Gatan imaging filter Tridiem and an EDS analyzer, was used to determine the structure and composition of the organo-mineral clusters that had accumulated on the surface of the aged biochar. Twenty biochar particles were sonicated in ethanol and a sample placed on a lacey carbon grind[35]. Detailed examination of two clusters was carried out using EELS and EDS[35].

## X-ray photoelectron spectroscopy (XPS)

XPS examination of fresh and 1.3-y aged (whole and crushed, <0.5 mm) biochar was undertaken in triplicate. The Carbon 1 s photoelectron peak was decomposed in five components, and allocated to bonds based on Singh et al.[40] (Supplementary Table 9).

## Liquid chromatography-organic carbon detection (LC-OCD)

Soil samples were extracted in distilled water (1:10 [w/v]) at 50 °C with regular stirring for 24 h before filtration to generate the liquid phase for analysis. Dissolved organic carbon (DOC) was analyzed using LC-OCD, yielding two major fractions: hydrophilic chromatographable organic carbon (CDOC) and hydrophobic organic carbon (HOC). CDOC was further categorized into five fractions based on retention time and molecular weight: biopolymers; persistent C-like substances; building blocks; low-molecular-weight (LMW) acids; and LMW neutrals. Quantification was performed using standards A (CDOC), B (HOC), C (biopolymers), D (persistent C), E (building blocks), and F (LMW neutrals).

Aromaticity was estimated using the ratio of spectral absorption coefficient measured for the persistent C substances normalized over the organic carbon value for persistent C substances.

## Calculations and statistical analyses

The cumulative SOC, biochar-C mineralization, and root respiration over 1.3 y (466 d) were calculated as the area of a linear interpolation across all measurement points. All statistical analyses were conducted within the R environment (R development core team, 2012). When F-tests were significant, means were separated using a least significant difference (LSD) test at $P = 0.05$. The first derivatives of the SXR spectra (281–296 eV) for each treatment were divided into 30 bins and pairwise correlations between each energy segment and distribution of aromatic-C, aliphatic-C, polysaccharide-C, clay-OH from IRM, and belowground $^{13}$C retention were assessed to explore relationships between C-speciation and C-distribution using R packages "prospectr" and "gplots".

## Scenario modelling

Global potential for wood biochar production is estimated at 0.31–0.59 Pg biochar annually, based on the total annual production of woody feedstock (i.e., forestry residues + agroforestry +green/wood waste) of 0.48–0.90 Pg C y$^{-1}$ under the 'beta' and 'Maximum Sustainable Technical Potential' (MSTP) scenarios, respectively, modelled by Woolf et al.[10], who assumed C yield of 49% (mass of C in the biochar divided by the mass of C in the initial dry biomass feedstock), and biochar C content of 75%. The alpha scenario assumes the conversion of biomass residues and wastes available using current technology and practices while the MSTP scenario assumes conversion of the maximum fraction of the global biomass resource that can be harvested without endangering food security, habitat, or soil conservation. At the rate applied in our study (10 Mg ha$^{-1}$), biochar could be applied to 31–59 Mha. We acknowledge that our result is likely to be specific to the context of this experiment, that is, Ferralsol under tropical grassland. Ferralsols occupy 750 Mha globally, almost exclusively in the tropics. Theoretically, all available biochar could be applied to similar sites globally. If the increase in soil C storage capacity observed in our study, 1.01 Mg SOC per Mg biochar-C from two applications of biochar,

was found to be a general response across similar sites, this could represent an additional soil C sink potential of 0.23–0.45 Pg C y$^{-1}$.

## Data availability

The authors declare that the $^{13}$C data of soil and CO$_2$ and results of belowground C allocation, XPS analysis on field-extracted biochar, aboveground biomass, microbial analyses, and processed synchrotron-based measurements supporting the findings of this study are available within the main text and its supplementary information files. The source data underlying Figs. 2, 4 and 5, as well as Supplementary Figs. 2–5, 7–9 generated in this study are provided in the Source Data file. Source data are provided in this paper. Source data are provided with this paper.

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

## Acknowledgements

The authors thank the Australian Government, Department of Agriculture and Water Resources for supporting the National Biochar Initiatives (2009–2012, 2012–2014; L.V.Z. and L.M.M.) that co-funded this research. Part of this research was undertaken on the SXR spectroscopy and the IR microspectroscopy beamlines at the Australian Synchrotron, part of ANSTO (Proposal IDs 15754 and 15940, Z.W. and P.M.K.). We thank the beamline scientists, Drs Bruce Cowie and Lars Thomsen, for their technical support on the SXR analysis. Part of the research is funded by La Trobe University's Research Focus Area in Securing Food, Water and the Environment (Grant Ready: SFWE RFA 2000004295; Z.W.; Collaboration Ready: SFWE RFA 2000004349; Z.W.). We appreciate the funding of Universities Australia and DAAD (Application ID: 57600933; Z.W., P.M.K., and S.S.) under the 2021 Australia-Germany Joint Research Co-operation Scheme for the development of image processing pipeline. We also appreciate the technical support from Mr Scott Petty and Mr Josh Rust for maintaining this field experiment over the past decade, and laboratory support from Ms Nichole Morris. We also thank Dr Carlos Achete from INMETRO (Brazil) and Dr Bin Gong from UNSW (Australia) for performing XPS analysis of biochars and soils, Dr Sarasadat Taherymoosavi from the University of New South Wales, Australia, for technical assistance in LC-OCD analysis. We acknowledge intellectual contributions from Dr Peter Slavich during manuscript preparation and Prof. Johannes Lehmann on the potential mechanisms of biochar-induced retention of rhizodeposits. We thank Mr Anders Claassens for graphics of Fig. 1 and Dr Natalie Betts for professional proofreading.

## Author contributions

Z.W. drafted and wrote the manuscript, designed and conducted experiments, and collected and analyzed data; L.V.Z., B.P.S., and L.M.M. wrote the manuscript, aided in experimental design, and provided critical revision of the article; S.S., A.R.K., J.V., M.J.T., S.J., E.T., B.S.A., H.D., S.L.Y.C., and M.T.R. collected and analyzed data, and provided critical revision of the article; T.J.R., C.T., A.F., P.M.K., S.K., S.M., and A.C. provided critical revision of the article. All authors have approved the final version for publication.

## Competing interests

The authors declare no competing interests.
