## [Peer Review File · Nature Communications]

Microspectroscopic visualization of how biochar lifts the soil organic carbon ceilingREVIEWER COMMENTS

Reviewer #1 (Remarks to the Author):

Dear Authors:

Reading the title, I thought that the contribution is certainly of interest for the broader audience and of high relevance to the field. Starting to read it, I realized that I had problems to follow, mainly because I could not follow the Materials and Methods part. It is written for specialists rather than for a common audience. However, being not able to understand the procedure in particular of the calculations and the approaches of the respiration studies, I realized I cannot judge the outcome of the paper. Considering that this may also be the case for other soil scientists not directly working in this field, I strongly recommend major revision with the goal to make it accessible for a broader audience. As it is now, the readers of nature communication, which is addressed to a broad and not specialist audience will be lost.

Reviewer #2 (Remarks to the Author):

The authors present an interesting hypothesis about the mechanisms by which biochar catalyzes the formation of new SOC and present total-C and ^{13}C isotopic data that support this hypothesis much as in their previous work (e.g., Weng et al. 2017 NatClimChange 7:371). What is new is the relatively rapid formation of additional SOC following a second amendment with the same biochar. They also present a large array of in-situ spectroscopic data that, while dazzling in its technical sophistication, does little to advance the hypothesis because there is no attempt to derive quantitative information from it. The reader is left feeling the job is only half completed. The XPS data for biochar particles is useful to show how biochar ages. The enzyme activity data probably suffer from artefactual issues related to different sorption affinities of the fluorophore and the analyte to biochar surfaces when compared to other soil surfaces (this potential error is not recognized by the general scientific community but needs to be investigated). Overall, the authors need to spend some time to further analyze the SXR, SIR, and EELS data for quantitative results that can help to prove the primary hypothesis. Figure 1 needs more work to fully represent the hypothesis proposed by the authors.

Specific comments

L 37: I think I know what an organo-mineral interface is, but it would help to define an organo-organic interface. Presumably organo refers to the sorbate and organic or mineral to the sorbent?

L 67-69: Is biochar-C considered part of SOC? If so, why is the initial increase in SOC after application of 7.6 Mg biochar-C/ha only 5 Mg C/ha? Or 6 Mg C/ha for the recent addition? Was some biochar C lost from the plots by erosion or drainage below the surface horizon? What is the measurement uncertainty?

L 119 ff: Enzyme activity—role of enzyme and MUB sorption to biochar? That is, how to distinguish between loss of measured enzyme activity due to MUB sorption vs. that due to enzyme sorption? Both likely occur and need to measure MUB sorption separately. The decrease in enzyme activity may simply be due to greater sorption of MUB by the biochar and thus an artifact.

L 131-132: “. . . formation of organo-mineral complexes . . .”

L 129-136: how representative are these analyses? Replicate samples analyzed or just one-offs?

L 193-194: Also should indicate that of the total increase of 24 Mg C/ha from historical/recent biochar additions, 15.2 Mg C/ha is from the biochar C and the other 8.8 Mg C/ha from the enhanced SOC stimulated by biochar. See also comment for L 67-69 above . . .

L 195ff: Somewhere in this paragraph the authors may want to consider citing Blanco-Canqui et al. 2019 (GCBBioenergy 12:240) for additional support of rapid stimulation of SOC formation/retention by biochar addition to soil.

L 228-229: Slavich et al. (2013) provide only the chemical properties of the biochar.

L 243: units for density are g cm⁻³. Quin et al. (2014) reference is not found in reference list.

L256-259: It would help to clarify whether the 13C data reported for each sampling are simply the 15-day data less those data taken immediately before the pulse-labeling expt. That is, do the total 13C capture data for the third sampling take into account the previous two pulse-labeling expts (3x190 = 570 mg 13C as the total or simply 190 mg 13C from the most recent pulse)?

L 269: Avoid use of “planted” and “unplanted” sub treatment descriptions? Use “soil+root” and “soil only” instead?

L 456-458: This section needs to be revised. Lehmann et al. (2011) do not provide any estimate of wood biochar production rate that I could find, and Jirka and Tomlinson (2013) reference is not provided. Perhaps the authors are thinking of Lehmann et al. (2006 MitigAdaptStratGlobChange 11:403), which is not cited and whose estimates (5.5-9.5 PgC/yr) may be considered overly optimistic. Woolf et al. (2010) estimate 0.9 Pg C per year as available woody feedstock in their maximum sustainable scenario (Table 1) and about half of this ends up as biochar(Figure 3). With this estimate there is still plenty of woody biochar available for the world's ferrasols (enough for 6-10 amendment cycles at 10 t BC carbon/ha).

L 547 and 570: Same reference listed twice.

Fig. 1: Hard to tell what is biochar in this figure (black background doesn't help!). The spawning of microaggregates from wet/dry cycles is not evident; Needs more work to bring out the concept fully.

Fig. 2 caption, L 479: do you mean 0-100 mm soil layer?

Fig. 3: What to “1.”, “2.”, and “3.” and associated arrows indicate in parts a) and b)? What does “19 +/- 2.3” mean and to what part(s) does it apply? 3c and 3d are technically wonderful, but don't really add

much to support the hypothesis—need some way of quantifying the effects, and of sorting out the biochar C associated with mineral surfaces from the SOC that might be forming on mineral surfaces. Were similar images to those in Fig. 3c,3d obtained for the control samples? Is the dense zone of C shown in Fig. 3c due to a biochar particle?

Fig. 4a: Vertical guidelines for i and ii are not parallel, tick marks on horizontal axis are not evenly spaced—it's hard to tell what energies are being displayed (change to primary tick marks for every 3-5 eV and secondary tick marks for intermediate integral energies). SXR energies for aggregates do not compare well with those for biochar (i.e., XPS in Table S9 and SXR in Fig. 5). Is this due to surface charging issues? Can you quantify the allocations to each type of carbon and compare with other estimates to derive a mechanistic description?

Fig. 4b: In the absence of any quantification, it's unclear what these images add to support the hypothesis. Are there biochar particles in the microaggregates shown in Fig. 4b?

Figure 5a: Use rational spacing for tick marks on horizontal axis (e.g., 0.5 eV or integral eV for minor tick marks)

Fig. 5 caption: description of part j) is not in figure, text seems related to Table S10 and Fig. S5 part i)?

Table S2: Analytical uncertainties not given

Table S5: Any idea why metabolic quotient of rhizodeposits is higher for recent+historical biochar treatment at 9.2 years but lower for the other sampling times?

Table S7: Why is the phosphatase activity lower with biochar? Sorption artifact?

Table S8: I am unsure what the significance is in comparison to. If zero, then why is not phosphatase in planted system significant at least for recent+historical biochar amendments?

Table S10: analytical uncertainties not given; LMW acids not listed, need to clarify what is meant by aromaticity listing; Difference between “planted” and “unplanted” treatments?

Fig S1: I would have expected the “planted” cumulative total flux to be higher than the “unplanted”. Any idea why they are essentially the same?

Fig S2: caption should refer to “squares” not “circles”

Fig S5: caption needed for images in section “i”

Reviewer #3 (Remarks to the Author):

The topic addressed in this manuscript is very important. The ability of biochar amendments to increase the carbon saturation level of soils (stimulate negative priming) has the potential to greatly enhance

both the agronomic and environmental benefits of biochar. If the negative priming effects of biochar can be managed at a field scale by farmers, then the amount of CO₂ removed from the atmosphere through biochar applications could increase substantially, which would make biochar one of, if not the, best approaches for mitigating climate change. Understanding the mechanisms and processes causing and controlling negative priming would therefore be an important scientific advance and could be a critical step towards developing practical tools for addressing climate change and simultaneously enhancing soil health and cropping systems resilience.

However, the manuscript, as currently written, is unacceptable. There are two major deficiencies in the manuscript; 1) the quality of the writing needs to be substantially improved, and 2) extrapolations from micron to plot scale and from plot scale to global scale are not scientifically justifiable.

The manuscript in general is very difficult to read because of the quality of the writing. Multiple topics and scales are jumbled into the same paragraph. A paragraph should begin with a topic sentence and focus on one and only one topic. Most paragraphs end with a concluding sentence and/or a sentence that transitions to the topic discussed in the succeeding paragraph. When paragraphs are composed of a jumble of topics, as in this paper, the poor reader will not be able to follow the logic. Furthermore, the entire manuscript is disorganized. New topics are discussed before the necessary context for the topics has been established. The sequence of paragraphs in a paper needs to be systematically aggregated so that each new paragraph builds on the previous paragraph and the totality tells a compelling story. The manuscript needs to be completely rewritten.

The data presented in Figure 2 is compelling and very valuable. However, the data presented in Figures 3, 4, and 5 has scale issues. The microscopic and spectroscopic data are valid for the specific micro aggregates analyzed. But there is no evidence that the selected microaggregates are representative of the soil in general or the plot. It is not appropriate to relate what is seen at the microaggregate scale to treatments which are imposed at the plot scale. Secondly, lines 203-208 extrapolate the observation for one plot study to the global potential of biochar to sequester carbon. The data to support such an extrapolation is not presented in this manuscript. To put this another way, you cannot extrapolate from what is essentially one data point to the entire planet.

NCOMMS-21-34507A Nature Communications review: “Microspectroscopic visualization of how biochar lifts the soil organic carbon ceiling”

Reviewers' comments:

Reviewer #1 (Remarks to the Author):

Dear Authors:

Reading the title, I thought that the contribution is certainly of interest for the broader audience and of high relevance to the field. Starting to read it, I realized that I had problems to follow, mainly because I could not follow the Materials and Methods part. It is written for specialists rather than for a common audience. However, being not able to understand the procedure in particular of the calculations and the approaches of the respiration studies, I realized I cannot judge the outcome of the paper. Considering that this may also be the case for other soil scientists not directly working in this field, I strongly recommend major revision with the goal to make it accessible for a broader audience. As it is now, the readers of nature communication, which is addressed to a broad and not specialist audience will be lost.

Authors' responses:

Thank you for helping us improving the manuscript for a broader audience. We have clarified the methods and reorganizing the sequences of the paragraphs for better logic flow. Particularly, to improve the readability of the complex methods of the respiration studies (*i.e.* priming), we have now created a diagram to summarize the calculations and approaches (Fig. S1). We have replaced acronym in the equations with full explanation to improve the clarity.

We have also reserved some technical aspect of the Materials and Methods for specialists who are interested to repeat the experiments and measurements.

For respiration, we have revised in Line 296: “Specialized respiration collars were used to isolate soil plus root respiration from shoot respiration^{41,42}. The visualization of the subplot set up and belowground respiration collars can be found in the supplementary information (Fig. S1). Moisture content was maintained between 60-80 % field capacity in the root collars to minimize potential C isotopic fractionation during photosynthesis caused by water stress⁴⁴.

Line 291: “A root signature sand collar (50 mm diameter) was installed in each of the control subplots to measure the $\delta^{13}C$ signature of root respiration. It was packed with acid-washed sand and planted with ryegrass (*i.e.* a down-sized version of the soil plus root respiration collar). Similarly, a biochar+root signature sand collar was packed with a biochar-sand mixture (1 % w/w) in each of the ‘recent + historical biochar’ subplots. To maintain the root growth into the collars, NPK fertilizers were applied at the same dose as in Slavich et al.⁴³.”

For calculations of the respiration studies, we have revised in Line 298: “To understand how plant-biochar-soil interactions affect SOC priming, $^{13}CO_2$ pulse labelling was implemented to the microplots to enrich the $\delta^{13}C$ signature of root (sand collars, Fig. S). Native SOC mineralization can then be separated from total respiration (soil+root collars) after pulse labelling. The pulse labelling campaigns were conducted on three occasions: 12 June 2014, 01 August 2014 and 30 July 2015. The same total amount of 190 mg $^{13}C\ m^{-2}$ was labelled and accounted for only for each labelling event. The

excess of enriched ^{13}C -CO₂ values of soil and root respiration relative to ^{13}C -CO₂ values of soil and root respiration at natural abundance in control and biochar-amended soils were measured after 3, 5, 10 and 15 days after 8.9-, 9.2-, and 9.5-year pulse labelling event. To avoid the disturbance to the microplots, the ^{13}C values for soil were only measured 15 d after each labelling event.

The rhizosphere priming of native SOC was quantified using a three-pool C partitioning model: (1) biochar-C, (2) root-C, and (3) SOC. The $\delta^{13}\text{C}$ signatures of extracted historical biochar and recent biochar (the same biochar archived in a sealed container for 8.2 years) were both -25 ± 0.1 ‰. Any interactive effect of biochar and root on the $\delta^{13}\text{C}$ signature of soil would be surpassed by a greater level of $\delta^{13}\text{C}$ enrichment of the root component compared with any isotopic signature contribution from soil and biochar to the $\delta^{13}\text{C}$ signature of the total respiration.

The mineralization of native SOC (C_{Soil}) was calculated using the ^{13}C signature of biochar+root ($^{13}\text{C}_{\text{Biochar+Root}}$, sand collar) from the soil+root systems after pulse labelling:

$$C_{\text{Soil}}(\%) = 100 \times (\delta^{13}\text{C}_{\text{Total}} - \delta^{13}\text{C}_{\text{Biochar+Root}}) / (\delta^{13}\text{C}_{\text{Soil}} - \delta^{13}\text{C}_{\text{Biochar+Root}}) \quad (1)$$

where $\delta^{13}\text{C}_{\text{Total}}$ is $\delta^{13}\text{C}$ signature of the total respiration from the soil+root collars after pulse labelling; $\delta^{13}\text{C}_{\text{Soil}}$ is $\delta^{13}\text{C}$ signature of the soil-derived CO₂-C evolved from the soil only control collars without pulse labelling.

Similarly, the percentage of soil-derived CO₂-C in the total respiration from the soil only biochar treatment (C_{Soil}' (%)) was determined:

$$C_{\text{Soil}}'(\%) = 100 \times (\delta^{13}\text{C}_{\text{Total}}' - \delta^{13}\text{C}_{\text{Biochar}}) / (\delta^{13}\text{C}_{\text{Soil}} - \delta^{13}\text{C}_{\text{Biochar}}) \quad (2)$$

Where $\delta^{13}\text{C}_{\text{Total}}'$ is the $\delta^{13}\text{C}$ signature of the total respiration from the soil only biochar treatment. $\delta^{13}\text{C}_{\text{Soil}}$ is the $\delta^{13}\text{C}$ signature of the soil-only control; $\delta^{13}\text{C}_{\text{Biochar}}$ is the $\delta^{13}\text{C}$ signature of either fresh (-25.02 ± 0.13 ‰) or aged biochar (-25.04 ± 0.11 ‰). Biochars were recovered by hand from field soil samples, thoroughly rinsed with distilled water on a 100 μm sieve and oven-dried at 50°C for 24 h.

Rhizosphere priming of SOC in the biochar system was the difference in native SOC mineralization between (1) the in the soil+root biochar treatment partitioned from ^{13}C -enriched 'biochar+root' endmember and (2) soil only biochar treatment partitioned from biochar end members:

$$\text{Priming} = (C_{\text{Soil}}(\%) \times C_{\text{Total}} - C_{\text{Soil}}'(\%) \times C_{\text{Total}}) / 100 \quad (3)''$$

Reviewer #2 (Remarks to the Author):

The authors present an interesting hypothesis about the mechanisms by which biochar catalyzes the formation of new SOC and present total-C and ¹³C isotopic data that support this hypothesis much as in their previous work (e.g., Weng et al. 2017 NatClimChange 7:371). What is new is the relatively rapid formation of additional SOC following a second amendment with the same biochar. They also present a large array of in-situ spectroscopic data that, while dazzling in its technical sophistication, does little to advance the hypothesis because there is no attempt to derive quantitative information from it. The reader is left feeling the job is only half completed. The XPS data for biochar particles is useful to show how biochar ages. The enzyme activity data probably suffer from artefactual issues related to different sorption affinities of the fluorophore and the analyte to biochar surfaces when compared to other soil surfaces (this potential error is not recognized by the general scientific community but needs to be investigated). Overall, the authors need to spend some time to further analyze the SXR, SIR, and EELS data for quantitative results that can help to prove the primary hypothesis. Figure 1 needs more work to fully represent the hypothesis proposed by the authors.

Authors' responses:

We would like to thank Reviewer #2 for recognizing the novelty of this current study. We have revised the manuscript thoroughly according to the reviewer's comments. We have also provided further analysis of the spectromicroscopic and enzyme activity data to support our hypothesis. We have improved Figure 1 to better illustrate the hypothesis.

Addressing the important concerns:

Firstly, we agree with Reviewer #2 that it is important to derive quantitative information of the *in situ* spectromicroscopic data. We integrated multi-scale and multi-dimensional techniques to better understand the relatively rapid formation of additional SOC following recent biochar application to a historical plot. To quantify direct visual evidence, we have developed an image processing pipeline for the IRM maps to calculate the distribution of C forms (Figs. 5b and S10). To link direct visual evidence with numerical measurements of soil carbon dynamics, we have further obtained first derivatives of the SXR spectra and conducted correlation analyses with IRM maps and belowground C retention (Fig. 5c). Quantitative EELS is challenging for carbon oxide radicals and is often qualitative in soil science. The limited option for quantitative EELS is often atoms ratio, such as C/O. However, we did not have data from oxygen. Because, these soil samples charged during the measurements, so collecting data twice (i.e. C and O) at the same place did not work for this circumstance. Nevertheless, we managed to integrate designated features in the C K edge spectra and created heat maps for indication of distribution of C functional groups at the nanometre scale (Fig. 4g & h).

Second, we acknowledge that Reviewer #2 has raised valid concerns regarding the measurements of enzyme activities. We have addressed these in detail below. In summary, (1) we have now recognized the potential sorption in the main text in Line 129: “It is noted that sorption affinities of the fluorophore and/or the enzyme to biochar compared to other soil surfaces may lead to underestimating the enzyme activities (Bailey et al. 2011).” Bailey, V. L., Fansler, S. J., Smith, J. L., & Bolton Jr, H. (2011). Reconciling apparent variability in effects of biochar amendment on soil enzyme activities by assay optimization. *Soil biology and biochemistry*, 43(2), 296-301; (2) we have clarified the original method, in particular, the use of actual samples (e.g. soils with or without biochar) to construct matrix-matched standard curves to account for potential sorption and/or quenching/excitation of the fluorophore; (3) we have further investigated this limitation of the enzyme assay with additional analyses and added in Line 132: “Here, we used matrix-matched standard curves to account for any potential binding (or quenching/excitation) of the fluorophore. The fluorescence response of standard curves constructed in soil matrix with or without biochar were not significantly different (Table S9), suggesting sorption, quenching or excitation were not important factors in the observed differences in enzyme activities.”

Fig. S10: Image processing pipeline for IRM maps.

Specific comments

L 37: I think I know what an organo-mineral interface is, but it would help to define an organo-organic interface. Presumably organo refers to the sorbate and organic or mineral to the sorbent?

Thank you for helping us improve the manuscript. We have now defined “organo-mineral interface” in Line 39: “Here, ‘organo’ refers to the sorbate and ‘organic’ or ‘mineral’ refers to the sorbent”.

L 67-69: Is biochar-C considered part of SOC? If so, why is the initial increase in SOC after application of 7.6 Mg biochar-C/ha only 5 Mg C/ha? Or 6 Mg C/ha for the recent addition? Was some biochar C lost from the plots by erosion or drainage below the surface horizon? What is the measurement uncertainty?

Thank you. Biochar-C was accounted in the total carbon stock. The variation in the difference between control and biochar plots (*i.e.* biochar-C) can be explained by the uncertainty of measurements. We have clarified the uncertainty in the measurements in Line 70: “We showed that a strategic application of 10 Mg biochar ha⁻¹ after 8.2 years raised the SOC storage ceiling by a further 2 Mg C ha⁻¹. Thus, this Ferralsol under the managed pasture had a C storage capacity of 35 (± 1.3) Mg C ha⁻¹ in the surface soil, which increased to 44 (± 0.7) Mg C ha⁻¹ one year following the application of biochar and reached 50 (± 1.1) Mg C ha⁻¹ after nearly a decade. The C storage ceiling was further raised to 58 (± 0.2) Mg C ha⁻¹, where biochar was applied to the historically amended field plots. Of this increase in SOC stock, 7.6 Mg C ha⁻¹ was derived from biochar, while 2 Mg C ha⁻¹ was attributed to the retention of new C. The total increase of 24 Mg C ha⁻¹ through the ‘recent + historical’ biochar amendment consisted of 15.2 Mg biochar-C ha⁻¹ and 8.8 Mg C ha⁻¹ from the biochar-enhanced SOC.”

L 119 ff: Enzyme activity—role of enzyme and MUB sorption to biochar? That is, how to distinguish between loss of measured enzyme activity due to MUB sorption vs. that due to enzyme sorption? Both likely occur and need to measure MUB sorption separately. The decrease in enzyme activity may simply be due to greater sorption of MUB by the biochar and thus an artifact.

This is an important point.

First, we have recognised the potential issues related to different sorption affinities of the fluorophore and the analyte to biochar surfaces when compared to other soil surfaces. We have added in Line 129: “It is noted that sorption affinities of the fluorophore and/or the enzyme to biochar compared to other soil surfaces may lead to underestimating the enzyme activities (Bailey et al. 2011)”.

Second, we have clarified the method for standard curves to minimise the impact of different sorption additivities. The fluorophore is MUB (product, not substrate). We used matrix-matched standard curves to account for any potential binding (or

quenching/excitation) of the fluorophore. This rules out under or overestimation of enzyme activities due to artefactual mis-quantification of the fluorophore. In any case, the fluorescence response of standard curves constructed in soil matrix with or without biochar were not significantly different (Table S9), suggesting sorption, quenching or excitation were not important factors in the observed differences in enzyme activities. Line 356: “Two standard curves were plotted for each measurement plate: one for soil (dilution series of 200 μ M 4-methylumbelliferyl (MUF) in 50 μ L of soil suspension from control or biochar amendments) and the other for hydrolysis (dilution series of 200 μ M MUF in 50 μ L of distilled water).”

Thirdly, we investigated the potential effects of substrate limitation through sorption processes. Previous research has suggested that biochar application at a rate of 3-15% can bind up to 30% of the substrate in colorimetry-based soil enzyme assays (Swaine et al. 2013). We tested the impact of potential substrate limitation by reducing the concentration of substrate in the assays by 50% (beyond the maximum reduction in substrate availability previously observed, at higher biochar rates). We found no significant difference in the rate of phosphatase activity when soil (no biochar) or soil (with biochar) was incubated with 1mM substrate (methylumbelliferyl phosphate) or with 0.5 mM substrate (data not shown). This indicates that even if sorption of substrate reached 50% of the amount supplied, it would not have affected the potential enzyme activity observed.

It remains unresolved as to whether reductions in potential enzyme activities in biochar-treated plots were due to reduced enzyme production, or due to increased binding/inactivation of enzymes by biochar (or biochar-mediated processes), or a combination of both. Nevertheless, our conclusions regarding the overall effect of biochar on soil enzyme activities remain valid since methodological artefacts were ruled out.

Swaine, M., Obrike, R., Clark, J. M., & Shaw, L. J. (2013). Biochar alteration of the sorption of substrates and products in soil enzyme assays. *Applied and Environmental Soil Science*, 2013.

L 131-132: “. . . formation of organo-mineral complexes . . .”

Thank you for spotting. We have revised Line 149: “These results from FIB-SEM-EDS illustrate how carbon (including rhizodeposits) can potentially be protected within soil by interacting with Fe oxides in the soil to promote the formation of organo-mineral complexes.”

L 129-136: how representative are these analyses? Replicate samples analyzed or just one-offs?

Thank you for helping us clarify the methods. We have replication of those spectromicroscopic analyses. This is one of the highlights of this current study. We emphasized in Line 139: “Replicated analyses were conducted to ensure the representative of direct visual evidence of the spatial heterogeneity at the nano- to micro-scales where the carbon accumulates”.

For SXR, Line 194: “From the SXR analyses (n =9), the C functional groups in the microaggregates (53-250 μm) were dominated by quinones (284.1 eV), aromatic C (285.2 eV, 1s- π^* transitions of conjugated C=C), and aliphatic C (287.3 eV).”

For IRM, detailed in Line 407: “For infrared microspectroscopy, approximately ~30 intact water-stable microaggregates (53–250 μm) and mineral fractions (<53 μm) were hand-picked on a glass fibre filter paper” and Line 419: “The IRM analysis was conducted using triplicate soil samples (i.e. three maps per treatment) resulting in a total of 815 individual spectrum measurements for microaggregates in the ‘recent + historical’ biochar treatment, 2335 spectra for mineral fraction in the ‘recent + historical’ biochar treatment, 1331 spectra for microaggregates in the ‘recent’ biochar treatment, and 1874 spectra for mineral fraction in the ‘recent’ biochar treatment.”

For EDS, Line 446: “Forty biochar particles were extracted from the soil samples per plot and were examined using a Zeiss Sigma scanning electron microscope. Detailed analysis of five particles was carried out using a Bruker X-ray Dispersive analyser (EDS).”

For EELS, Line 451: “Twenty biochar particles were sonicated in ethanol and then a sample of this was placed on a lacey carbon grid as described by Archanjo et al (2017). Detailed examination of two clusters was carried out using energy electron loss spectroscopy (EELS) and EDS.”

For XPS, Line 454: “X-ray photoelectron spectroscopy (XPS) examination of both whole and crushed (< 0.5 mm) 1-year aged particles of biochar was undertaken in triplicate.”

L 193-194: Also should indicate that of the total increase of 24 Mg C/ha from historical/recent biochar additions, 15.2 Mg C/ha is from the biochar C and the other 8.8 Mg C/ha from the enhanced SOC stimulated by biochar. See also comment for L 67-69 above . . .

Thank you for the suggestions. We have added in Line 76: “The total increase of 24 Mg C ha⁻¹ through the ‘recent + historical’ biochar amendment consisted of 15.2 Mg biochar-C ha⁻¹ and 8.8 Mg C ha⁻¹ from the biochar-enhanced SOC”.

L 195ff: Somewhere in this paragraph the authors may want to consider citing Blanco-Canqui et al. 2019 (GCBioenergy 12:240) for additional support of rapid stimulation of SOC formation/retention by biochar addition to soil.

Thank you. We have added in Line 198: “The elevation of the SOC ceiling observed in our trial has significant implications for the global efforts to build SOC^{9, 60, 61}”.

60. Schmidt, M. W. *et al.* Persistence of soil organic matter as an ecosystem property. *Nature* **478**, 49-56 (2011).

61. Blanco-Canqui, H., Laird, D. A., Heaton, E. A., Rathke, S., & Acharya, B. S. Soil carbon increased by twice the amount of biochar carbon applied after 6 years: Field evidence of negative priming. *GCB Bioenergy* **12**, 240-251 (2020).

L 228-229: Slavich et al. (2013) provide only the chemical properties of the biochar.

We have revised Line 247: “The chemical properties of the biochar can be found in Slavich et al. (2013).”

L 243: units for density are g cm⁻³. Quin et al. (2014) reference is not found in reference list.

We have revised the unit for density. Line 260: “The bulk density of the biochar was 0.332 g cm⁻³ measured using a method described in Quin et al. (2014).” We have also added the reference: “Quin, P. R. et al. Oil mallee biochar improves soil structural properties—A study with x-ray micro-CT. *Agriculture, ecosystems & environment* **191**, 142-149 (2014).”

L256-259: It would help to clarify whether the 13C data reported for each sampling are simply the 15-day data less those data taken immediately before the pulse-labeling expt. That is, do the total 13C capture data for the third sampling take into account the previous two pulse-labeling expts (3x190 = 570 mg 13C as the total or simply 190 mg 13C from the most recent pulse)?

Thank you for helping us improve the clarity of the method. We have clarified in Line 312: “The same total amount of 190 mg ¹³C m⁻² was labelled and accounted for only for each labelling event. The excess of enriched ¹³C-CO₂ values of soil and root respiration relative to ¹³C-CO₂ values of soil and root respiration at natural abundance in control and biochar-amended soils were measured after 3, 5, 10 and 15 days after 8.9-, 9.2-, and 9.5-year pulse labelling event. To avoid the disturbance to the microplots, the ¹³C values for soil were only measured 15 d after each labelling event.”

L 269: Avoid use of “planted” and “unplanted” sub treatment descriptions? Use “soil+root” and “soil only” instead?

Thank you for the suggestion. We have replaced “planted” and “unplanted” with “soil+root” and “soil only” throughout the manuscript. For example, Line 278: “At the 15-month soil sampling event, fresh soil was also taken from the soil respiration collars (i.e. soil only) and soil+root respiration collars to quantify the effect of plant-biochar-soil interactions”. Accordingly in the Figures.

L 456-458: This section needs to be revised. Lehmann et al. (2011) do not provide any estimate of wood biochar production rate that I could find, and Jirka and Tomlinson (2013) reference is not provided. Perhaps the authors are thinking of Lehmann et al. (2006 MitigAdaptStratGlobChange 11:403), which is not cited and whose estimates (5.5-9.5 PgC/yr) may be considered overly optimistic. Woolf et al. (2010) estimate 0.9 Pg C per year as available woody feedstock in their maximum sustainable scenario (Table 1) and about

half of this ends up as biochar(Figure 3). With this estimate there is still plenty of woody biochar available for the world's ferrasols (enough for 6-10 amendment cycles at 10 t BC carbon/ha).

Thank you for helping us improve this important section. We have revised this section based on the suggestion in Line 486: "Global potential for wood biochar production is estimated at 0.3-0.6 Pg, based on the total annual production of woody feedstock of 0.48-0.90 Pg C yr⁻¹ under the 'alpha' and 'Maximum Sustainable Technical Potential' (MSTP) scenarios, respectively, modeled by Woolf et al¹⁰, who assumed C yield of 49% (mass of C in the biochar divided by the mass of C in the initial dry biomass feedstock), and biochar C content of 75%".

L 547 and 570: Same reference listed twice.

We have removed the duplication.

Fig. 1: Hard to tell what is biochar in this figure (black background doesn't help!). The spawning of microaggregates from wet/dry cycles is not evident; Needs more work to bring out the concept fully.

Thank you for your suggestions. We have improved Figure 1 with lighter colour to increase contrast between biochar particles and this dark red Ferrasol. We have removed the spawning of microaggregates from wet/dry cycles. We have clarified the stepwise hypotheses in the new Figure 1.

Fig. 2 caption, L 479: do you mean 0-100 mm soil layer?

We have clarified in the method in Line 272: “Intact soil cores (40 mm in diameter) were sampled to 80 mm depth within each subplot” and Line 282: “The total SOC content was converted to soil carbon density in the 0-80 mm soil the using bulk density assessed in each biochar treatment.”

Fig. 3: What to “1.”, “2.”, and “3.” and associated arrows indicate in parts a) and b)? What does “19 +/- 2.3” mean and to what part(s) does it apply? 3c and 3d are technically wonderful, but don’t really add much to support the hypothesis—need some way of quantifying the effects, and of sorting out the biochar C associated with mineral surfaces from the SOC that might be forming on mineral surfaces. Were similar images to those in Fig. 3c,3d obtained for the control samples? Is the dense zone of C shown in Fig. 3c due to a biochar particle?

We have removed the numbering in the Figure 3. We have also removed “19 ± 2.3” in the key to avoid confusion.

We have clarified the importance and relevance of the first use of 3D FIB-SEM-EDX on intact soil aggregates to the hypothesis in Line 144: “To visualize the elemental composition of intact aggregates, three-dimensional (3D) distribution of C, Si, Al, and Fe were assessed using focused ion beam (FIB) coupled with scanning electron microscopy (SEM) with energy-dispersive X-ray spectroscopy (FIB-SEM-EDS). Co-location of carbon with Si-, Al- and Fe-rich clay minerals was observed in the core of aggregates from the ‘recent + historical’ biochar treatment (Fig. 3c), but not in the ‘recent’ biochar treatment (Fig. 3d). These results from FIB-SEM-EDS illustrate how carbon (including rhizodeposits) can potentially be protected within soil by interacting with Fe oxides in the soil to promote the formation of organo-mineral complexes”. We have also conducted FIB-SEM-EDX on the control treatment (Fig. S6i).

Although FIB-SEM-EDX only showed total C distribution, the dense zone of C in Figure 3 was likely from rhizodeposits as revealed by sequential SEM images of the development of a root channel (Supplementary video).

We agree with Reviewer #2 that it is important to quantify the effects and differentiated various C sources. Here, we quantified the belowground ¹³C retention and partitioned three C sources (i.e. soil, biochar and root-derived C) in Line 86: “To further examine this decrease in SOC mineralization, we partitioned rhizodeposits (root C) from biochar C and SOC through aggregate size and density fractionation”. We have provided quantitative analyses of spectromicroscopic data detailed in the next response. We developed an image processing pipeline for IRM maps to quantify the microscale distribution of C forms associated with clay minerals, Line 208: “To quantify these observations from IRM, we developed an image processing pipeline to quantify the distribution of C forms in association with clay (Fig. S10). For the microaggregates, greater aromatic C (31% of pixels across the intact section) was found in the ‘recent + historical’ biochar compared with the ‘recent’ biochar treatment (22% of pixels) because of the recent biochar dose (Fig. 5b). The distribution of polysaccharide C (36-42%), aromatic C (17-19%), and clay (12-14%) was similar in the mineral fraction of

the 'recent' and 'recent + historical' biochar treatments. Greater aliphatic C (33%) was observed in the mineral fraction of the 'recent+ historical' than the 'recent' biochar treatment (27%). These observations are in agreement with new C retention in belowground ¹³C pools (Figs. 3a, 3b & 5c), highlighting the importance of clay minerals for protecting SOC from microbial mineralization.”

Fig. 4a: Vertical guidelines for i and ii are not parallel, tick marks on horizontal axis are not evenly spaced—it's hard to tell what energies are being displayed (change to primary tick marks for every 3-5 eV and secondary tick marks for intermediate integral energies). SXR energies for aggregates do not compare well with those for biochar(i.e., XPS in Table S9 and SXR in Fig. 5). Is this due to surface charging issues? Can you quantify the allocations to each type of carbon and compare with other estimates to derive a mechanistic description?

Thank you for the suggestions. We have improved the display of Fig. 4a with new tick marks (3 eV for primary tick marks and 0.5 eV for secondary tick marks). SXR energies were the same for aggregates and biochars. We have improved both figures to make them more comparable as suggested. There was surface charging during the measurement. Flood gun was applied to both samples, Line 403: “Flood gun was used to due to the surface charging of the samples”. For the quantification of SXR spectra, we have now calculated the first derivatives and conducted correlative analyses to link spectroscopic data with belowground ¹³C retention and distribution of C forms in IRM maps (Fig. 5c).

Fig. 4b: In the absence of any quantification, it's unclear what these images add to support the hypothesis. Are there biochar particles in the microaggregates shown in Fig. 4b?

We have provided quantification of the IRM maps by conducting correlation analysis and developing an image processing pipeline and have indicated that high intensity of aromatic C may serve as tracer for biochar particles. We have added in Line 201: These SXR results also align with the micro-spatial maps produced from the IRM analyses of intact sections taken from microaggregates and mineral fraction (Fig.5c), with high intensity of aromatic C as a tracer for biochar. We found that the correlation between clay minerals and microbial metabolites (aliphatic C) in the macroaggregates in the 'recent + historical' biochar treatment ($R^2 = 0.96$, Fig. S9) was stronger than in the recent biochar treatment ($R^2 = 0.86$). In contrast, the distribution of microbial-derived C with clay is similar for the mineral fraction ($R^2 = 0.94$). The correlation between polysaccharide C and clay was greater in the macroaggregates of the 'recent + historical' biochar treatment ($R^2 = 0.83$) than for the 'recent' biochar treatment ($R^2 = 0.46$). To quantify these observations from IRM, we developed an image processing pipeline to quantify the distribution of C forms in association with clay (Fig. S10). For the microaggregates, greater aromatic C (31% of pixels across the intact section) was found in the 'recent + historical' biochar compared with the 'recent' biochar treatment (22% of pixels) because of the recent biochar dose (Fig. 5b). The distribution of

polysaccharide C (36-42%), aromatic C (17-19%), and clay (12-14%) was similar in the mineral fraction of the 'recent' and 'recent + historical' biochar treatments. Greater aliphatic C (33%) was observed in the mineral fraction of the 'recent+ historical' than the 'recent' biochar treatment (27%). These observations are in agreement with new C retention in belowground ¹³C pools (Figs. 3a, 3b & 5c), highlighting the importance of clay minerals for protecting SOC from microbial mineralization."

In methods, Line 419: "The IRM analysis was conducted using triplicate soil samples (i.e. three maps per treatment) resulting in a total of 815 individual spectrum measurements for microaggregates in the 'recent + historical' biochar treatment, 2335 spectra for mineral fraction in the 'recent + historical' biochar treatment, 1331 spectra for microaggregates in the 'recent' biochar treatment, and 1874 spectra for mineral fraction in the 'recent' biochar treatment.

Figure 5a: Use rational spacing for tick marks on horizontal axis (e.g., 0.5 eV or integral eV for minor tick marks)

We have revised the minor tick marks to 0.5 eV on horizontal axis.

Fig. 5 caption: description of part j) is not in figure, text seems related to Table S10 and Fig. S5 part i)?

We have included the description in now Fig. 4j.

Table S2: Analytical uncertainties not given

We have provided analytical uncertainties.

Table S5: Any idea why metabolic quotient of rhizodeposits is higher for recent+historical biochar treatment at 9.2 years but lower for the other sampling times?

This is likely due to increase of root-derived respiration in the "recent+historical" biochar treatment at 9.2 years ($0.52 \mu\text{g CO}_2\text{-C mg}^{-1}$) compared with the 'recent' biochar treatment ($0.19 \mu\text{g CO}_2\text{-C mg}^{-1}$) whilst MBC remained similar ($1.45 \text{ cf. } 1.33 \text{ mg MCB g}^{-1} \text{ soil}$).

Table S7: Why is the phosphatase activity lower with biochar? Sorption artifact?

There is a possibility due to high variability of the enzyme assay for the phosphatase activity, given control of 609 ± 129 , recent biochar of 418 ± 52 , and recent + historical biochar of $322 \pm 58 \text{ nmol MUB mg}^{-1} \text{ MBC h}^{-1}$.

Table S8: I am unsure what the significance is in comparison to. If zero, then

why is not phosphatase in planted system significant at least for recent+historical biochar amendments?

Thank you for pointing this out. We have revised the table.

Table S10: analytical uncertainties not given; LMW acids not listed, need to clarify what is meant by aromaticity listing; Difference between “planted” and “unplanted” treatments?

We have now provided analytical uncertainties for the LC-OCD analysis and clarified aromaticity. LMW acids was not quantified. We added in the caption: “Aromaticity is estimated by the ratio of spectral absorption coefficient measured for the persistent C substances normalised over the organic carbon value for persistent C substances”. Only ‘soil+root’ samples (planted) were analysed.

Fig S1: I would have expected the “planted” cumulative total flux to be higher than the “unplanted”. Any idea why they are essentially the same?

That is correct. We have now kept the vertical axis consistent for both figures.

Fig S2: caption should refer to “squares” not “circles”

Thank you for spotting this. Fig S2 caption now reads: “Cumulative root respiration for all treatments. Controls contained no biochar (open squares). Recent biochar is the first application of biochar in the unamended control soil (dark squares). The recent + historical biochar is fresh biochar application to the 9.5-year field-aged biochar-amended soil (dashed squares). Confidence intervals (95%) (n= 3) are plotted using dashed lines for the controls and normalised against the mean squares across all treatments. The six arrows represent N fertiliser addition”.

Fig S5: caption needed for images in section “I”

We have removed Fig. S5i.

Reviewer #3 (Remarks to the Author):

The topic addressed in this manuscript is very important. The ability of biochar amendments to increase the carbon saturation level of soils (stimulate negative priming) has the potential to greatly enhance both the agronomic and environmental benefits of biochar. If the negative priming effects of biochar can be managed at a field scale by farmers, then the amount of CO₂ removed from the atmosphere through biochar applications could increase substantially, which would make biochar one of, if not the, best approaches for mitigating climate change. Understanding the mechanisms and processes causing and controlling negative priming would therefore be an important scientific advance and could be a critical step towards developing practical tools for addressing climate change and simultaneously enhancing soil health and cropping systems resilience.

However, the manuscript, as currently written, is unacceptable. There are two major deficiencies in the manuscript; 1) the quality of the writing needs to be substantially improved, and 2) extrapolations from micron to plot scale and from plot scale to global scale are not scientifically justifiable.

The manuscript in general is very difficult to read because of the quality of the writing. Multiple topics and scales are jumbled into the same paragraph. A paragraph should begin with a topic sentence and focus on one and only one topic. Most paragraphs end with a concluding sentence and/or a sentence that transitions to the topic discussed in the succeeding paragraph. When paragraphs are composed of a jumble of topics, as in this paper, the poor reader will not be able to follow the logic. Furthermore, the entire manuscript is disorganized. New topics are discussed before the necessary context for the topics has been established. The sequence of paragraphs in a paper needs to be systematically aggregated so that each new paragraph builds on the previous paragraph and the totality tells a compelling story. The manuscript needs to be completely rewritten.

The data presented in Figure 2 is compelling and very valuable. However, the data presented in Figures 3, 4, and 5 has scale issues. The microscopic and spectroscopic data are valid for the specific micro aggregates analyzed. But there is no evidence that the selected microaggregates are representative of the soil in general or the plot. It is not appropriate to relate what is seen at the microaggregate scale to treatments which are imposed at the plot scale. Secondly, lines 203-208 extrapolate the observation for one plot study to the global potential of biochar to sequester carbon. The data to support such an extrapolation is not presented in this manuscript. To put this another way, you cannot extrapolate from what is essentially one data point to the entire planet.

Thank you for recognizing the importance of our work.

Addressing the major concerns:

- 1) We have improved the quality of the writing. We have strengthened the logic flow by reorganizing the paragraphs. Within each paragraph, we simplify to one topic per paragraph with a topic sentence and a concluding sentence. For the introduction, for example, we outline: (i) SOC has been lost; (ii) Humanity now faces the grand challenge to urgently reduce CO₂ emissions, including the ongoing loss of SOC from soil; (iii) the IPCC has identified that substantial CDR will be needed to meet the Paris Agreement target of limiting warming to well below 2deg; (iv) IPCC has identified soil C management and biochar as CDR methods with considerable potential; (v) biochar's capacity to protect and build SOC, which is often overlooked, could raise the potential for CO₂ removal through soil C management (hypothesis paragraph with Fig. 1). We then discussed the results in a logic order, "Lifting storage capacity of soil organic carbon": (i) total SOC, (ii) priming, and (iii) belowground C retention; "Microbial contribution and responses to the retention of rhizodeposits": (i) microbial biomass, (ii) substrate included respiration, and (iii) enzyme activities; "Spatial heterogeneity of soil organic carbon": (i) 3D elemental distribution of intact soil aggregates, (ii) 2D spectromicroscopic changes in C functional groups on biochar surfaces at the nanoscale, (iii) 2D microspectroscopic changes in C forms and distribution in soil at the microscale with quantification; global implications of our findings.
- 2) We have clarified that we did not intend to extrapolate data from a single field study to the global scale. Instead, we aim to convey the potential significance of our results with global implications justified with comprehensive modelling from the literature. We have now corrected the source of the biochar production potential. We have clarified in the main text in Line 66: "Our results showed that the SOC storage ceiling could be lifted through single or multiple applications of biochars. We observed a plateau in rhizodeposit accumulation rate over 9.5 years in the historical biochar plots (Fig. 2a; $y = 4.24\ln(x) + 17.6$; $R^2 = 0.95$), implying that the system was approaching a new (16 % higher) equilibrium for SOC storage, ten years after the initial application. We showed that a strategic application of 10 Mg biochar ha⁻¹ after 8.2 years raised the SOC storage ceiling by a further 2 Mg C ha⁻¹. Thus, this Ferralsol under the managed pasture had a C storage capacity of 35 (± 1.3) Mg C ha⁻¹ in the surface soil, which increased to 44 (± 0.7) Mg C ha⁻¹ one year following the application of biochar and reached 50 (± 1.1) Mg C ha⁻¹ after nearly a decade. The C storage ceiling was further raised to 58 (± 0.2) Mg C ha⁻¹, where biochar was applied to the historically amended field plots. Of this increase in SOC stock, 7.6 Mg C ha⁻¹ was derived from biochar, while 2 Mg C ha⁻¹ was attributed to the retention of new C. The total increase of 24 Mg C ha⁻¹ through the 'recent + historical' biochar amendment consisted of 15.2 Mg biochar-C ha⁻¹ and 8.8 Mg C ha⁻¹ from the biochar-enhanced SOC."

Line 219: "The elevation of the SOC ceiling observed in our trial has significant implications for the global efforts to build SOC^{9, 60, 61}. Plants release ~50 % of photosynthetically-fixed C into the soil, which supports microbial

growth⁵⁷⁻⁵⁹. Grasslands annually constitute 0.04 Pg C to the global SOC pool⁶. To demonstrate the magnitude of the potential CDR through raising the C storage ceiling, we estimated the soil carbon increase that could be delivered if the increase in soil carbon storage capacity observed in this study was found to be a general response across similar sites. Based on global potential production of woody biochar of 0.48-0.90 Pg C yr⁻¹ (see Methods), assuming this biochar is applied to Ferralsols under tropical pasture and the same response of 0.5 Mg SOC per Mg biochar applied was achieved, this could represent an additional soil C sink potential of 0.12-0.22 Pg C.”

We further clarified in the methods in Line 486: “Global potential for wood biochar production is estimated at 0.3-0.6 Pg, based on the total annual production of woody feedstock of 0.48-0.90 Pg C yr⁻¹ under the ‘alpha’ and ‘Maximum Sustainable Technical Potential’ (MSTP) scenarios, respectively, modeled by Woolf et al¹⁰, who assumed C yield of 49% (mass of C in the biochar divided by the mass of C in the initial dry biomass feedstock), and biochar C content of 75%. The alpha scenario assumes the conversion of biomass residues and wastes available using current technology and practices while the MSTP scenario assumes conversion of the maximum fraction of the global biomass resource that can be harvested without endangering food security, habitat or soil conservation. At the rate applied in our study, 10 Mg ha⁻¹, biochar could be applied to 30-60 M ha. We acknowledge that our result is likely to be specific to the context of this experiment, that is, Ferralsol soil type under tropical grassland. Ferralsols occupy 750 M ha globally, almost exclusively in the tropics. Thus, theoretically, biochar could be applied to similar sites globally. If the increase in soil carbon storage capacity observed in our study, 0.5 Mg SOC per Mg biochar applied, was found to be a general response across similar sites, this could represent an additional soil C sink potential of 0.12-0.22 Pg C.”

REVIEWER COMMENTS

Reviewer #2 (Remarks to the Author):

The authors have made many improvements to the manuscript. Nevertheless, the manuscript suffers from numerous instances where the textual discussion seems somewhat divorced from the values presented in the figures and tables. At the very least, the reader will have trouble understanding how the values used in the discussion are derived. These instances are listed below followed by a list of minor edits.

1) L 68-75: The text in lines 68-75 still does not seem internally consistent with respect to C stocks. You start with 35 Mg C ha⁻¹, add 10 Mg ha⁻¹ of biochar(76% C = 7.6 Mg C ha⁻¹) twice and that gives you about 50.2 Mg C ha⁻¹ without counting any new npSOC. After 9.5 years you report 58 Mg C ha⁻¹ as the total increase (23 Mg C ha⁻¹) but then in L 75-76 discuss the increase as being 24 Mg C ha⁻¹, of which 15.2 is biochar C and 8.8 is npSOC. In L 72-73, you also discuss the increase to 50 in historical plots and 58 in historical+recent plots, a difference of 8 Mg C ha⁻¹, then in L 73-74 you allocate this to 7.6 Mg C ha⁻¹ of biochar C and 2 Mg C ha⁻¹ of npSOC, which is a total of 9.6 Mg C ha⁻¹ rather than 8 Mg C ha⁻¹ in the previous sentence. This is all confusing to the reader and needs to be cleaned up so that all numbers reported are consistent and the readers can follow the changes without scratching their head. One thing that would help is to specify the sampling time exactly (e.g., 9.5 y) rather than talking about “after nearly a decade” and “further raising it to . . .”.

2) L 85-88: I am having trouble reproducing the 16% increase in C retention mentioned in L 85-88 and attributed to Figure 3a. Please clarify how the values in Figure 3a lead to the values in the text. Perhaps Figure S4 is more relevant, although it is difficult for the reader to derive accurate numbers from the bar chart.

3) L 67: When I plug x=10 into the equation given for Figure 2a ($y = 4.24\ln(x) + 17.6$) I get a value of 27.3. This doesn't seem to match values of total soil C for historical plots in the figure. Some work is needed here to help the reader.

4) L 68: New equilibrium in Fig 2a seems to be around 52 Mg ha⁻¹, vs. about 40-42 originally if one doesn't include the historical biochar C added. The increase thus seems to be 24% to 30% depending on which initial values are selected, not 16%. If one starts with the original soil C level of 35 Mg C ha⁻¹ and adds 16 Mg C ha⁻¹ then one is close to the values plotted. So maybe you mean to say 16 Mg C ha⁻¹ rather than 16%?

5) L 104-105: The data in Table S5a show that microbial biomass in “recent+historical” biochar is less than “recent” biochar by 10%. Another mismatch between text and data. If the data are correct, this will also change the discussion in L 106-108.

6) L 112: I agree that for bulk SOC it is as you state. For rhizodeposition, however, this does not seem to be the case. The data are either not statistically significant or, as for the 9.2-year results, just the opposite.

7) L 121-123: The differences in Table S7 do not seem to be statistically significant (eyeballing the SDs) except possibly for Xylosidase and Phosphatase. On the other hand, many of the enzyme activities do seem to be statistically different from the control (Table S8). This seems to be the opposite of what is stated in the text.

Minor Edits:

Figure 1 is much improved!

L 20-21: consider adding recent IPCC AR6WGIII report on mitigation (<https://www.ipcc.ch/report/sixth-assessment-report-working-group-3/>) as reference.

L 56-57: change to read “. . . biochar. To quantify increased C storage capacity . . .”

Figure 2a: Why are there only three triangles shown for the “recent+historical” biochar doses when there are four data points for each of the “recent” and “historical” biochar doses? Unless the dark triangle at about 50 Mg ha⁻¹ is the “recent+historical” data point right on top of one of the “historical” data points.

L 131-133: Glad to see that this potential source of error was investigated and seems to be negligible for this system!

L 137-139: The last sentence in the paragraph does not make sense and needs work.

L 143-145: change to read “aggregates, the three-dimensional . . .(SEM) and elemental detection provided by energy-dispersive . . .”

L 156-157: Delete “It is known that”

L 165-167: Can you provide integrated peak areas (or plot overlapping spectra) to back up this assertion? The eye struggles to see much difference between the two patterns as currently shown.

L 185: Do you mean “validated”?

Figure 5a: It would help to provide a table with deconvoluted peak areas for the specific spectral classes in the SXR data.

L 208-213, Figure 5b and Table S12: May want to provide a statistical analysis for the functional groups shown in the pie charts and table? It would help the discussion to identify where the significant differences are among the 4 classes of samples analyzed.

L 230-231: How could one “accelerate . . . heterogeneity and temporal variability”? As written, this is an awkward, almost meaningless sentence.

Figure S9 caption: Do you mean to refer to Figure 5 in the main text? Which axis is for clay (x?) and which for the carbon spectral classes (y?)? Also, change last sentence to read “. . . absorption peak could be detected above baseline noise . . .”.

L 238: change to read “. . . the soil is a Rhodic Ferrasol, . . .”

L 272: “. . . using a vortex mixer) . . .”

L 299-300: Consider this phrasing if accurate: “Each event applied 190 mg $^{13}\text{C m}^{-2}$ as the label and was analyzed as an independent experiment assuming no retention of ^{13}C from prior events.”

L 325: delete “the in”

L 382: “. . . in an FEI SCIOS . . .”

L 400-401: “. . . An electron flood gun was used to minimize surface charging . . .”

L 408: “No embedding medium was used.”

Reviewer #3 (Remarks to the Author):

The manuscript confirms numerous previous findings that biochar can induce negative priming. The first contribution of this study (Figure 2) is evidence that negative priming persists for many years but appears to approach a new plateau (new saturation level) after 9.5 years for the studied Ferrasol. Figure 2 also shows that second biochar addition 8.2 years into the study can re-initiate negative priming (further increase the C saturation level). Figure 2 data is derived from microplots (440 mm diameter) with biochar applied in the top 10 cm. The second contribution (Figures 3, 4, & 5) is substantial spectroscopic data at the molecular to microaggregate scales showing accumulation of clay mineral-organic complexes in the soil. This spectroscopic evidence supports a model (Figure 1) for one possible mechanism by which biochar might promote accumulation of new biogenic soil organic carbon.

The writing in the revised manuscript is substantially improved over the first manuscript. But is still challenging in places for a general reader to decipher.

My second major concern related to extrapolation from the plot scale to the global scale and from nano scale to the plot scale. The authors have revised their discussion of quantitative extrapolation (L271-225) from the plot scale to the global scale. They state that the purpose of the exercise is only to show that biochar induced negative priming has potential to make a major contribution towards CDR and acknowledge that the exercise is highly context dependent and not a precise estimate of global CDR potential. The authors, however, did not address the second half of this concern (extrapolation from the molecular/microaggregate scales to the plot scale). Specifically in Figures 3, 4, & 5 there is no evidence that the samples analyzed at the molecular/microaggregate scales are representative of the soils that received the recent and recent + historical treatments at the plot scale. My recommendation is to use the spectroscopic data to show the formation of clay-organic complexes and to support the model (Figure 1) as one possible mechanism by which biochar promotes the accumulation of new biogenic SOC. But you should acknowledge that other mechanisms are also possible.

One weakness in the model (Figure 1) is that newly formed clay-organic complexes are visualized as sluffing off of biochar surfaces to make room for new clay-organic complexes to form. I see no evidence to support “sluffing off”? Perhaps the surfaces and pore spaces require ~8 years to be saturated.

Other comments:

L106: “might result” sounds like speculation.

L111: “may partially explain” sounds like speculation.

L113: “The ‘recent + historical’ biochar might result in higher substrate-use efficiency which supports an earlier negative priming than the ‘recent’ biochar (Fig. 2b).” “might result in higher” can be stated more succinctly as “increased”. The word “might” implies that this is speculation rather than an evidence based statement. Speculation is not evidence that “supports an earlier negative priming...”

L123: “This suggests that for a given amount of microbial biomass, less enzymes were produced in the biochar-amended soil.” “suggests” sounds like speculation. I suggest that you present ‘more enzymes were adsorbed’ (L128-129) as an alternate explanation, then state your reasons (L131-133) for thinking that it is less important.

L135: Microbial-driven retention of SOC does not make sense. Microbes want to use SOC as a substrate not store it for C sequestration. Also, ‘rhizodeposits’ are a form of ‘SOC’. This sentence implies that they are two different materials.

L137-139: “replicate analyses” of a few nano-grams of soil does not “ensure” that the samples are representative of anything.

L147-149: The results may illustrate a possible mechanism; but co-location of C with Si, Al, & Fe in the recent + historical samples and not in the recent samples (L146-147) – is meaningless. You only examined a few nano-grams of soil, not enough to imply a process difference between treatments that were imposed at the micro-plot scale.

L160-164: How do you know that the mineral surfaces were positively charged and the biochar surfaces were negatively charged? Fe-Al-oxyhydroxide minerals can have both positive and negative net surface charge depending on pH. Most biochars are dominated by neutral carboxyl or negative carboxylate groups – depending on pH, but some biochars may also have positively charged oxonium groups.

L165-166: “Specifically, using synchrotron-based soft X-ray (SXR), we observed greater intensities of quinones (284.1 eV) and carboxyl C–OOH (288.6 eV) in the 9.5-year aged biochar compared with the 1-year aged biochar (Fig. 167 4i).” Analyzing a few nano-grams of soil is not sufficient evidence to support a process difference between 9.5 year aged and 1-year aged treatments imposed at the plot scale.

L184: “...appear to be influenced by...” again this sounds like speculation rather than hard evidence.

L191-192: “Protection of rhizodeposits from microbial metabolism is evident from several observations.” These are interesting observations (in this paragraph) but how do they prove “protection”?

L201-206: These correlations provide some quantitative evidence. But it is not clear how many observations (n) were used? And were these observations all within the same microaggregate? Again, there is a risk of sampling bias. One microaggregate or even a few microaggregates does not represent a significant amount of soil at the plot scale where recent and recent + historical treatments were imposed.

L229-231: “Our in situ spectromicroscopic analyses suggest that the catalytic biochar surfaces accelerated the micro- and nano-scale heterogeneity and temporal variability for new C storage.” The meaning of this sentence is not clear. Biochar might accelerate the rate of C storage; but how can it accelerate “heterogeneity” and “temporal variability”? Heterogeneity of surface sites might increase when you add biochar to a soil but heterogeneity is not going to “accelerate”. “Temporal variability” of what? Even if biochar increases the heterogeneity of surface sites (which means increases the number of different types of surface sites), how does this increase the storage of new C? Explaining the mechanism by which biochar increases storage of new C is potentially the critical scientific contribution of this paper, however, this sentence fails to achieve that critical objective. I suggest that you describe the model (Figure 1) in some detail along with reference back to the spectroscopic evidence that supports it. Then acknowledge that this is only one possible mechanism.

NCOMMS-21-34507B Nature Communications review: “Microspectroscopic visualization of how biochar lifts the soil organic carbon ceiling”

REVIEWER COMMENTS

Reviewer #2 (Remarks to the Author):

The authors have made many improvements to the manuscript. Nevertheless, the manuscript suffers from numerous instances where the textual discussion seems somewhat divorced from the values presented in the figures and tables. At the very least, the reader will have trouble understanding how the values used in the discussion are derived. These instances are listed below followed by a list of minor edits.

We thank the reviewer for providing insightful comments to improve our manuscript. We have now cross checked and synchronised the textual discussion and values in the display items. The revised manuscript has also been reviewed by professional proofreading service to improve the clarity and readability to the wide readership of Nature Communications. Please find point-by-point responses below. All line numbers correspond to the clean version of the revised manuscript with reviewers' specific comments highlighted in green (Reviewer #2) and yellow (Reviewer #3).

1) L 68-75: The text in lines 68-75 still does not seem internally consistent with respect to C stocks. You start with 35 Mg C ha⁻¹, add 10 Mg ha⁻¹ of biochar (76% C = 7.6 Mg BCC ha⁻¹) twice and that gives you about 50.2 Mg C ha⁻¹ without counting any new npSOC. After 9.5 years you report 58 Mg C ha⁻¹ as the total increase (23 Mg C ha⁻¹) but then in L 75-76 discuss the increase as being 24 Mg C ha⁻¹, of which 15.2 is biochar C and 8.8 is npSOC. In L 72-73, you also discuss the increase to 50 in historical plots and 58 in historical+recent plots, a difference of 8 Mg C ha⁻¹, then in L 73-74 you allocate this to 7.6 Mg C ha⁻¹ of biochar C and 2 Mg C ha⁻¹ of npSOC, which is a total of 9.6 Mg C ha⁻¹ rather than 8 Mg C ha⁻¹ in the previous sentence. This is all confusing to the reader and needs to be cleaned up so that all numbers reported are consistent and the readers can follow the changes without scratching their head. One thing that would help is to specify the sampling time exactly (e.g., 9.5 y) rather than talking about “after nearly a decade” and “further raising it to . . .”.

Thank you for helping us convey this important message clearer. We have revised in Lines 58-71: “Our results showed that the SOC storage ceiling could be lifted through either single or multiple applications of biochar (Fig. 2a). The Control stored 35 (\pm 1.3) Mg C ha⁻¹ in topsoil (0-75mm), while the Historical plots stored 50 (\pm 1.1) Mg C ha⁻¹ at 9.5 y after biochar addition (Fig. 2a). When biochar was added to the Control plots after 8.2 y (Control+Recent), the SOC storage capacity was raised to 44 (\pm 0.7) Mg C ha⁻¹ 1.3 y following biochar application, while a second application of biochar after 8.2 years (Historical+Recent) raised the total SOC to 58 (\pm 0.2) Mg C

ha⁻¹. The total increase of 15 Mg C ha⁻¹ after 9.5 y in Historical soils consisted of 5.7 Mg biochar-C ha⁻¹ in the top 75mm layer and 9.3 Mg C ha⁻¹ from the enhanced SOC accumulation. However, this enhanced SOC accumulation could be increased by multiple applications of biochar – the total increase of 23 Mg C ha⁻¹ in the Historical+Recent treatment after 1.3 y consisted of 11.4 Mg biochar-C ha⁻¹ and 11.6 Mg C ha⁻¹ from enhanced SOC accumulation. Thus, the second application of biochar in the Historical+Recent soil increased the SOC storage capacity by an additional 2.3 Mg new C ha⁻¹ compared to the Historical soil with a single application of biochar, with this being a 25% increase in new SOC accumulation caused by the second application of biochar”. We have further clarified the amount of biochar-C in the method section in Line 272: “Note that although 7.6 Mg biochar-C ha⁻¹ was incorporated to 100 mm depth, soils were sampled to 75mm depth because the trial originally started as an ‘agronomic assessment of biochar’ and the industry standard for pasture soil analysis was 0-75mm sampling. Hence, the amount of biochar-C in the top 75mm layer was estimated to be 5.7 Mg biochar-C ha⁻¹ assuming no lateral movement of biochar. This may underestimate new SOC accumulation”.

2) L 85-88: I am having trouble reproducing the 16% increase in C retention mentioned in L 85-88 and attributed to Figure 3a. Please clarify how the values in Figure 3a lead to the values in the text. Perhaps Figure S4 is more relevant, although it is difficult for the reader to derive accurate numbers from the bar chart.

Thank you for pointing out. We have clarified the 16% was drawn between the total SOC stock of Historical+Recent (58 Mg ha⁻¹) vs Historical (50 Mg ha⁻¹) treatments. However, the novelty of this manuscript draws on the further accumulation of new SOC with multiple biochar applications. Therefore, we have now removed this sentence. We have revised in Line 68: “the second application of biochar in the Historical+Recent soil increased the SOC storage capacity by an additional 2.3 Mg new C ha⁻¹ compared to the Historical soil with a single application of biochar, with this being a 25% increase in new SOC accumulation caused by the second application of biochar”.

3) L 67: When I plug $x=10$ into the equation given for Figure 2a ($y = 4.24\ln(x) + 17.6$) I get a value of 27.3. This doesn't seem to match values of total soil C for historical plots in the figure. Some work is needed here to help the reader.

Thank you. The original equation was based on sampling dates. As suggested by Reviewer #3 regarding extrapolation from the plot scale to the global scale plots, we have now removed this equation to avoid speculation.

4) L 68: New equilibrium in Fig 2a seems to be around 52 Mg ha⁻¹, vs. about 40-42 originally if one doesn't include the historical biochar C added. The increase thus seems to be 24% to 30% depending on which initial values are selected, not 16%. If one starts with the original soil C level of 35 Mg C ha⁻¹

and adds 16 Mg C ha⁻¹ then one is close to the values plotted. So maybe you mean to say 16 Mg C ha⁻¹ rather than 16%?

We have revised accordingly as detailed above.

5) L 104-105: The data in Table S5a show that microbial biomass in “recent+historical” biochar is less than “recent” biochar by 10%. Another mismatch between text and data. If the data are correct, this will also change the discussion in L 106-108.

Thank you for spotting this. We have revised Line 95: “Microbial biomass increased by 8-12% in Control+Recent compared with Historical+Recent soils between 8.9–9.5 years (Table S5), likely due to the stimulation of microbial co-metabolism³¹ by the addition of biochar-C to a previously unamended soil, which also induced a small positive priming effect in Control+Recent soils (Fig. 2b)”.

6) L 112: I agree that for bulk SOC it is as you state. For rhizodeposition, however, this does not seem to be the case. The data are either not statistically significant or, as for the 9.2-year results, just the opposite.

We have revised Line 101: “This greater respiration induced by carboxylic and phenolic acids (typically in root exudates) partially explained the higher metabolic quotient associated with bulk SOC in Control+Recent vs Historical+Recent soils (Table S5)” and in Line 114: “This is consistent with decreased metabolic quotients/increased microbial C-use efficiency (Table S5) for bulk SOC but not root-derived C in the amended soils, which indirectly contributes to negative priming (Fig. 2b)”.

7) L 121-123: The differences in Table S7 do not seem to be statistically significant (eyeballing the SDs) except possibly for Xylosidase and Phosphatase. On the other hand, many of the enzyme activities do seem to be statistically different from the control (Table S8). This seems to be the opposite of what is stated in the text.

We have double checked and revised Line 112: “Here, the ratio of enzyme activity-to-microbial biomass was similar in both Historical+Recent and Control+Recent soils compared to the Control (Table S6) despite reduced enzyme activities (Table S7)”.

Minor Edits:

Figure 1 is much improved!

Thank you.

L 20-21: consider adding recent IPCC AR6WGIII report on mitigation (<https://www.ipcc.ch/report/sixth-assessment-report-working-group-3/>) as reference.

We have now included this reference: IPCC, 2022: Climate Change 2022: Mitigation of Climate Change. Contribution of Working Group III to the Sixth Assessment Report of the Intergovernmental Panel on Climate Change [P.R. Shukla, J. Skea, R. Slade, A. Al Khourdajie, R. van Diemen, D. McCollum, M. Pathak, S. Some, P. Vyas, R. Fradera, M. Belkacemi, A. Hasija, G. Lisboa, S. Luz, J. Malley, (eds.)]. Cambridge University Press, Cambridge, UK and New York, NY, USA. doi: 10.1017/9781009157926

L 56-57: change to read “. . . biochar. To quantify increased C storage capacity . . .”

We have revised Line 49: “To examine the potential for biochar to protect soil organic matter from microbial degradation, we measured SOC stocks in a biochar-amended managed pasture over 9.5 years.”

Figure 2a: Why are there only three triangles shown for the “recent+historical” biochar doses when there are four data points for each of the “recent” and “historical” biochar doses? Unless the dark triangle at about 50 Mg ha⁻¹ is the “recent+historical” data point right on top of one of the “historical” data points.

Thank you for pointing this out. There are four triangles for the Historical+Recent treatment, but, the data points overlapped with the Historical treatment.

L 131-133: Glad to see that this potential source of error was investigated and seems to be negligible for this system!

Thank you for pointing out this potential limitation of the enzyme assay which deserves recognition by the general scientific community.

L 137-139: The last sentence in the paragraph does not make sense and needs work.

We have revised this sentence based on the comments from Reviewers #2 and #3 in Line 136: “To visualize the retention of rhizodeposits and microbial-derived C, we undertook one-dimensional (1D) spectroscopic, 2D microspectroscopic, and 3D

electron microscopic analyses of SOC spatial heterogeneity. We provided direct visual evidence of the spatial heterogeneity at the nano- to micro-scales”.

L 143-145: change to read “aggregates, the three-dimensional . . .(SEM) and elemental detection provided by energy-dispersive . . .”

Thank you. We have revised Line 139: “To better understand the process of negative priming following biochar application, we mapped the elemental composition within intact aggregates to determine whether the retention of rhizodeposits (and other forms of C) may be facilitated via protection by Fe and Al-rich soil minerals. The 3D distribution of C, Si, Al, and Fe was assessed using a focused ion beam (FIB) coupled with scanning electron microscopy (SEM) and elemental detection provided by energy-dispersive X-ray spectroscopy (FIB-SEM-EDS; Fig. 3c, 3d)”.

L 156-157: Delete “It is known that”

Deleted. Line 149 now reads: “Fungi can mine nutrients from minerals by exuding acids^{47,48} that may cause the observed microporosity of organo–mineral–biochar interfaces (Figs. 4a–c, e, S6).

L 165-167: Can you provide integrated peak areas (or plot overlapping spectra) to back up this assertion? The eye struggles to see much difference between the two patterns as currently shown.

We now provide deconvolution and integration of peak areas in Table S11. We have revised Line 160: “Using synchrotron-based soft X-ray (SXR) spectroscopy (Fig. 4i), we observed greater intensities of carboxyl C–OOH (288.6 eV) in the 9.5-year aged biochar (10.6%) compared with the 1-year aged biochar (6.1%; Table S11)”.

L 185: Do you mean “validated”?

Yes. We have revised in Line 184: “We further validated the nanoscale observations of biochar surfaces in soil at the microscale”.

Figure 5a: It would help to provide a table with deconvoluted peak areas for the specific spectral classes in the SXR data.

A table of deconvoluted peak areas (Table S11) is added to the supplementary information.

L 208-213, Figure 5b and Table S12: May want to provide a statistical analysis for the functional groups shown in the pie charts and table? It would help the discussion to identify where the significant differences are among the 4 classes of samples analyzed.

Thank you for the suggestion. We have provided statistical analyses of the functional groups in Table S12. We have revised Line 209: “For the microaggregates, a greater proportion of aromatic-C (31% of pixels across the intact section) was found in Historical+Recent soil compared with Control+Recent soil (22%) because of the biochar persisting in Historical soils from the original soil amendment at trial establishment (Fig. 5b; Table S12). The distribution of polysaccharide-C (36–42%), aromatic-C (17–19%), aliphatic C (27–33%), and clay (12–14%) was similar in the two mineral fractions (Fig. 5b; Table S12)”.

L 230-231: How could one “accelerate . . . heterogeneity and temporal variability”? As written, this is an awkward, almost meaningless sentence.

Thank you. We have revised this sentence and also incorporated Reviewer #3’s comment in Line 230: “Our in situ spectromicroscopic analyses at the molecular to microaggregate scales showed accumulation of clay mineral-organic complexes in the soil”.

Figure S9 caption: Do you mean to refer to Figure 5 in the main text? Which axis is for clay (x?) and which for the carbon spectral classes (y?)? Also, change last sentence to read “. . . absorption peak could be detected above baseline noise . . .”.

Thank you. We have revised accordingly.

L 238: change to read “. . . the soil is a Rhodic Ferrasol, . . .”

We have revised in Line 239: “Briefly, the soil is a Rhodic Ferralsol, a fine-textured and Fe-rich mineral soil dominated by kaolinite, gibbsite and goethite”.

L 272: “. . . using a vortex mixer) . . .”

We have revised in Line 279: “Soil pH was measured on soil suspensions (1:5 w/w soil:water; Weng et al., 2017)”.

L 299-300: Consider this phrasing if accurate: “Each event applied 190 mg ¹³C m⁻² as the label and was analyzed as an independent experiment assuming no retention of ¹³C from prior events.”

Thank you for helping us improve the clarity of the sentence. We have revised Line 291: “Each event applied 190 mg ¹³C m⁻² as the label and was analyzed as an independent experiment assuming no retention of ¹³C from prior events.”.

L 325: delete “the in”

Deleted.

L 382: “. . . in an FEI SCIOS . . .”

We have revised Line 365: “Soil particle sections for EDS mapping were prepared in an FEI SCIOS FIB/SEM DualBeam system”.

L 400-401: “. . . An electron flood gun was used to minimize surface charging . . .”

We have revised Line 379: “An electron flood gun was used to minimize surface charging”.

L 408: “No embedding medium was used.”

We have revised Line 391: “No embedding medium was used.”

Reviewer #3 (Remarks to the Author):

The manuscript confirms numerous previous findings that biochar can induce negative priming. The first contribution of this study (Figure 2) is evidence that negative priming persists for many years but appears to approach a new plateau (new saturation level) after 9.5 years for the studied Ferrasol. Figure 2 also shows that second biochar addition 8.2 years into the study can re-initiate negative priming (further increase the C saturation level). Figure 2 data is derived from microplots (440 mm diameter) with biochar applied in the top 10 cm. The second contribution (Figures 3, 4, & 5) is substantial spectroscopic data at the molecular to microaggregate scales showing accumulation of clay mineral-organic complexes in the soil. This spectroscopic evidence supports a model (Figure 1) for one possible mechanism by which biochar might promote accumulation of new biogenic soil organic carbon.

We thank the reviewer for recognising the improvement of our manuscript based on all reviewers' comments. We believe that we have now resolved all two issues the reviewer raised. Please find point-by-point responses below. All line numbers correspond to the clean version of the revised manuscript with reviewers' specific comments highlighted in green (Reviewer #2) and yellow (Reviewer #3).

The writing in the revised manuscript is substantially improved over the first manuscript. But is still challenging in places for a general reader to decipher.

Firstly, we have obtained professional proofreading to further improve the clarity of the manuscript.

My second major concern related to extrapolation from the plot scale to the global scale and from nano scale to the plot scale. The authors have revised their discussion of quantitative extrapolation (L271-225) from the plot scale to the global scale. They state that the purpose of the exercise is only to show that biochar induced negative priming has potential to make a major contribution towards CDR and acknowledge that the exercise is highly context dependent and not a precise estimate of global CDR potential. The authors, however, did not address the second half of this concern (extrapolation from the molecular/microaggregate scales to the plot scale). Specifically in Figures 3, 4, & 5 there is no evidence that the samples analyzed at the molecular/microaggregate scales are representative of the soils that received the recent and recent + historical treatments at the plot scale. My recommendation is to use the spectroscopic data to show the formation of clay-organic complexes and to support the model (Figure 1) as one possible mechanism by which biochar promotes the accumulation of new biogenic SOC. But you should acknowledge that other mechanisms are also possible.

One weakness in the model (Figure 1) is that newly formed clay-organic complexes are visualized as sluffing off of biochar surfaces to make room for new clay-organic complexes to form. I see no evidence to support “sluffing off”? Perhaps the surfaces and pore spaces require ~8 years to be saturated.

Secondly, we agree with the reviewer regarding the challenge of extrapolation from molecular/microaggregate to plot scale. We now address this issue by adapting the reviewer’s recommendation to use spectromicroscopic data to support the model in Figure 1. We clarify that this is principally used as a piece of supporting evidence and we do not make definitive statements about the implications of this in terms of processes at the plot scale. We also remove the “sluffing off” process in the model. We have revised Figure 1 accordingly to reflect this change.

We have restructured the results and discussion section on the molecular/microaggregate data, starting in Line 126: “Our study provides the first visual evidence of a mechanism by which biochar can accelerate the formation of organo-mineral and organic interfaces in soils to protect SOC from microbial degradation, summarized in Fig. 1. Biochar can sorb root-derived C (rhizodeposits) that forms biofilms on its surfaces (Figs. 1a and 3a). The very fine layer of soil minerals that accumulate on biochar as it ages in soil³⁰⁻³² protects rhizodeposits from microbial metabolism^{33,34} over time, microbial necromass is also incorporated into this coating of organo-mineral and organic interfaces and is protected from degradation³⁵⁻⁴⁰ (Figs. 1b,c, 4, and 5). A coating can build on the biochar surfaces (Fig. 1d) and the processes repeat to build rhizodeposits in soil over time (Fig. 1e). Our spectroscopic data showed the formation of clay–organic complexes as one possible mechanism by which biochar promotes the accumulation of new biogenic SOC”.

Other comments:

L106: “might result” sounds like speculation.

We have revised Line 95: “Microbial biomass increased by 8-12% in Control+Recent compared with Historical+Recent soils between 8.9–9.5 years (Table S5), as a result of the stimulation of microbial co-metabolism³¹ by the addition of biochar-C to a previously unamended soil, which also induced a small positive priming effect in Control+Recent soils (Fig. 2b)”.

L111: “may partially explain” sounds like speculation.

We have revised Line 101: “This greater respiration induced by carboxylic and phenolic acids (typically in root exudates) partially explained the higher metabolic quotient associated with bulk SOC in Control+Recent vs Historical+Recent soils (Table S5)”.

L113: “The ‘recent + historical’ biochar might result in higher substrate-use efficiency which supports an earlier negative priming than the ‘recent’ biochar (Fig. 2b).” “might result in higher” can be stated more succinctly as “increased”. The word “might” implies that this is speculation rather than an evidence based statement. Speculation is not evidence that “supports an earlier negative priming...”

Thank you. We have revised Line 104: “Lower metabolic quotients indicate higher substrate-use efficiency, so the lower metabolic quotient observed in Historical+Recent soils supports the more rapid establishment of negative priming than the Control+Recent soils after the biochar addition at 8.2 y (Fig. 2b)”.

L123: “This suggests that for a given amount of microbial biomass, less enzymes were produced in the biochar-amended soil.” “suggests” sounds like speculation. I suggest that you present ‘more enzymes were adsorbed’ (L128-129) as an alternate explanation, then state your reasons (L131-133) for thinking that it is less important.

Thank you. We have revised this section based on the suggestion in Line 112: “Here, the ratio of enzyme activity-to-microbial biomass was similar in both Historical+Recent and Control+Recent soils compared to the Control (Table S6) despite reduced enzyme activities (Table S7). This is consistent with decreased metabolic quotients/increased microbial C-use efficiency (Table S5) for bulk SOC but not root-derived C in the amended soils, which indirectly contributes to negative priming (Fig. 2b). The presence of opportunistic microbes that meet their energy and nutrient demands by exploiting the catalytic activities of decomposers could lower the specific enzyme activity⁴⁵. It is noted that sorption affinities of the fluorophore and/or the enzyme to biochar compared to other soil surfaces may lead to underestimating enzyme activities⁴⁶. Here, we used matrix-matched standard curves

to account for any potential binding (or quenching/excitation) of the fluorophore. The fluorescence response of standard curves constructed using the soil matrix with or without biochar were not significantly different (Table S8), suggesting that fluorophore sorption, quenching, or excitation did not contribute to the observed differences in enzyme activities”.

L135: Microbial-driven retention of SOC does not make sense. Microbes want to use SOC as a substrate not store it for C sequestration. Also, ‘rhizodeposits’ are a form of ‘SOC’. This sentence implies that they are two different materials.

Thank you for pointing this out. We have revised Line 136: “To visualize the retention of rhizodeposits and microbial-derived C, we undertook one-dimensional (1D) spectroscopic, 2D microspectroscopic, and 3D electron microscopic analyses of SOC spatial heterogeneity”.

L137-139: “replicate analyses” of a few nano-grams of soil does not “ensure” that the samples are representative of anything.

Thank you. We agree. As suggested by the reviewer, here we present spectromicroscopic data as supporting evidence to quantification of SOC stock and dynamics through stable isotopic approaches. We have taken all possible steps to ensure that the sample analysed is representative. Composite samples were collected within each field replicate and further laboratory replicates were applied. We have reworded in Line 138: “We provided direct visual evidence of the spatial heterogeneity at the nano- to micro-scales”.

L147-149: The results may illustrate a possible mechanism; but co-location of C with Si, Al, & Fe in the recent + historical samples and not in the recent samples (L146-147) – is meaningless. You only examined a few nano-grams of soil, not enough to imply a process difference between treatments that were imposed at the micro-plot scale.

Agreed. We have removed this sentence.

L160-164: How do you know that the mineral surfaces were positively charged and the biochar surfaces were negatively charged? Fe-Al-oxyhydroxide minerals can have both positive and negative net surface charge depending on pH. Most biochars are dominated by neutral carboxyl or negative carboxylate groups – depending on pH, but some biochars may also have positively charged oxonium groups.

Thank you for the comment. We have explained that this is one possible and acknowledged other possibility in Line 157: “It is also noted that Fe-Al-oxyhydroxide minerals can have both positive and negative net surface charge depending on pH (*i.e.*, they are variable charge minerals). Most biochars are dominated by neutral

carboxyl or negative carboxylate group (depending on pH), but some biochars may also have positively charged oxonium groups”.

L165-166: “Specifically, using synchrotron-based soft X-ray (SXR), we observed greater intensities of quinones (284.1 eV) and carboxyl C–OOH (288.6 eV) in the 9.5-year aged biochar compared with the 1-year aged biochar(Fig. 167 4i).” Analyzing a few nano-grams of soil is not sufficient evidence to support a process difference between 9.5 year aged and 1-year aged treatments imposed at the plot scale.

This is a challenge that soil scientists have faced for the last century. Even in ‘routine’ soil chemical analyses, if approximately 1 g of soil is analysed, this only represents 0.00000077% of the soil in 1 ha to 10 cm depth. Thus, the challenge highlighted here by the reviewer is not a challenge that is unique for our study. Regardless, we have taken all possible steps to ensure that the sample analysed is representative. Composite samples were collected within each field replicate and further laboratory replicates were applied. In addition, we have modified the manuscript in Line 160: “Using synchrotron-based soft X-ray (SXR) spectroscopy (Fig. 4i), we observed greater intensities of carboxyl C–OOH (288.6 eV) in the 9.5-year aged biochar (10.6%) compared with the 1-year aged biochar (6.1%; Table S11). Similarly, exudates from plants and microorganisms can be deposited around minerals and attracted by cations onto biochar surfaces. Recent biochar amendment to the Historical plots would provide new unoccupied surfaces and pores in the soil to increase sorption capacity for root exudates⁴⁹ (Fig. 1a), which would then serve as binding agents to further enhance aggregation⁵⁰. As these clusters build, they may also be detached from the biochar either through fluctuating redox conditions, interaction with microbes, or perturbation caused by soil invertebrates or human activities³⁰ (Fig. 1d). These results provide direct evidence of repeated cycles of formation of organo-mineral coatings on the biochar surfaces during aggregate turnover or in response to changes in soil conditions, with these processes accumulating rhizodeposits in soil over time”. This is principally used as a piece of supporting evidence and we do not make definitive statements about the implications of this in terms of processes at the plot scale”. We clarified the sample number in the method section: “Composite of five samples were collected in each field replicate and three laboratory replicates were obtained for each of three field replicates (n=9).”

L184: “...appear to be influenced by...” again this sounds like speculation rather than hard evidence.

We have revised Line 183: “The concentrations of the different functional groups were influenced by the presence of nanophase Fe, Si and Al oxides⁵⁴”.

L191-192: “Protection of rhizodeposits from microbial metabolism is evident from several observations.” These are interesting observation (in this paragraph) but how do they prove “protection”?

We have provided further interpretation of the observations in Line 196: “These data provide evidence of rhizodeposit and microbial necromass incorporation into SOC, with rhizodeposits predominantly in microaggregates rather than mineral fractions. This difference indicates that retention of rhizodeposits in SOC relies on forming complex organic and organo-mineral interfaces with microbial necromass and biochar, while microbial necromass can be retained by organo-mineral interfaces in mineral fractions”.

L201-206: These correlations provide some quantitative evidence. But it is not clear how many observations (n) were used? And were these observations all within the same microaggregate? Again, there is a risk of sampling bias. One microaggregate or even a few microaggregates does not represent a significant amount of soil at the plot scale where recent and recent + historical treatments were imposed.

We have clarified the correlations in Line 398: “The IRM analysis was conducted using triplicate soil samples (*i.e.* three maps per treatment) resulting in a total of 815 individual spectrum measurements for microaggregates in the Historical+Recent treatment, 2335 spectra for mineral fraction in the Historical+Recent treatment, 1331 spectra for microaggregates in the Control+Recent treatment, and 1874 spectra for mineral fraction in the Control+Recent treatment”.

L229-231: “Our in situ spectromicroscopic analyses suggest that the catalytic biochar surfaces accelerated the micro- and nano-scale heterogeneity and temporal variability for new C storage.” The meaning of this sentence is not clear. Biochar might accelerate the rate of C storage; but how can it accelerate “heterogeneity” and “temporal variability”? Heterogeneity of surface site might increase when you add biochar to a soil but heterogeneity is not going to “accelerate”. “Temporal variability” of what? Even if biochar increases the heterogeneity of surface sites (which means increases the number of different types of surface site), how does this increase the storage of new C? Explaining the mechanism by which biochar increases storage of new C is potentially the critical scientific contribution of this paper, however, this sentence fails to achieve that critical objective. I suggest that you describe the model (Figure 1) in some detail along with reference back to the spectroscopic evidence that supports it. Then acknowledge that this is only one possible mechanism.

Thank you. We have restructured the discussion around the model as suggested by the Reviewer in Line 126: ‘Our study provides the first visual evidence of a mechanism by which biochar can accelerate the formation of organo-mineral and organic interfaces in soils to protect SOC from microbial degradation, summarized in Fig. 1. Biochar can sorb root-derived C (rhizodeposits) that forms biofilms on its surfaces (Figs. 1a and 3a). The very fine layer of soil minerals that accumulate on biochar as it ages in soil³⁰⁻³² protects rhizodeposits from microbial metabolism^{33,34} over time, microbial necromass is also incorporated into this coating of organo-mineral and organic interfaces and is protected from degradation³⁵⁻⁴⁰ (Figs. 1b,c, 4,

and 5). A coating can build on the biochar surfaces (Fig. 1d) and the processes repeat to build rhizodeposits in soil over time (Fig. 1e). Our spectroscopic data showed the formation of clay–organic complexes as one possible mechanism by which biochar promotes the accumulation of new biogenic SOC”.

We have clarified in Line 230: “Our *in situ* spectromicroscopic analyses at the molecular to microaggregate scales showed accumulation of clay mineral-organic complexes in the soil. This spectroscopic evidence supports our proposed model (Fig. 1) for one possible mechanism by which biochar promotes accumulation of new biogenic SOC. This mechanism, if found to apply in other tropical Ferralsols, could substantially increase the potential of biochar as a CDR.”

REVIEWERS' COMMENTS

Reviewer #3 (Remarks to the Author):

The manuscript is an important contribution that furthers understanding of negative priming induced by biochar and the potential of biochar to impact CDR and soil quality/health. The authors have adequately addressed concerns raised during the review process. And I am now pleased to recommend publication of the manuscript.

As the authors rightly point out the soil sampling problem (extrapolating from observations made on nano- and micro-grams samples to processes occurring at the (mega- and giga-gram scale) plot and field scales) is endemic to soil science in general and not unique to this paper.

My congratulations to the authors.

NCOMMS-21-34507C Nature Communications review: “Microspectroscopic visualization of how biochar lifts the soil organic carbon ceiling”

REVIEWER COMMENTS

Reviewer #3 (Remarks to the Author):

The manuscript is an important contribution that furthers understanding of negative priming induced by biochar and the potential of biochar to impact CDR and soil quality/health. The authors have adequately addressed concerns raised during the review process. And I am now pleased to recommend publication of the manuscript.

As the authors rightly point out the soil sampling problem (extrapolating from observations made on nano- and micro-grams samples to processes occurring at the (mega- and giga-gram scale) plot and field scales) is endemic to soil science in general and not unique to this paper.

My congratulations to the authors.

We thank the reviewer for providing insightful comments through the previous revisions. We are pleased to hear your recommendation. We appreciate the Reviewers' time and effort in helping us to improve the manuscript and to ensure that it has an important impact in the scientific literature.